# Full Implementation of the River Chief System in China: Outcome and Weakness

**Yinghong Li** [1], **Jiaxin Tong** [2] **and Longfei Wang** [2,*]

1   School of Marxism, Hohai University, Nanjing 210098, China; yingshanhongli@163.com
2   Key Laboratory of Integrated Regulation and Resource Development on Shallow Lakes,
    Ministry of Education, College of Environment, Hohai University, Nanjing 210098, China;
    frogtong01@gmail.com
*   Correspondence: lfwang@hhu.edu.cn

**Abstract:** Despite having explored various modes of water management over the past three decades, the water crisis persists and the Chinese government has been required to revolutionize river management from the top down. The River Chief System (RCS), which evolved from small scale, local efforts to manage rivers starting in 2007, is an innovative system that coordinates between existing 'fragmented' river/lake management and pollution control systems, to clearly define the responsibilities of all concerned departments. The system was promoted from an emergent policy to nationwide action in 2016, and ever since, has undergone steady development. We have analyzed recent developments in the system from the perspectives of functional expansion, implementation strategies, legislative processes, and public outreach after the full implementation of the RCS. By collecting data over the past several years, the changes in the water quality of representative watersheds in China were evaluated to assess the outcomes of RCS implementation. Finally, a summary of the weaknesses and outstanding problems of the system is presented, putting forward a multi-channel strategy for the long-term stability and effectiveness of river/lake chiefs, and promoting the RCS as a suitable solution to the collaborative and jurisdictional issues in water management in China.

**Keywords:** river chief system; lake chief system; water management; legislation and regulation; implementation strategy

## 1. Introduction

Collaborative governance is a core tenet of public management [1]. The management of rivers and lakes is a typical problem dealt with by public affairs, it is complex, highly variable, and has a high degree of uncertainty [2]. There are challenging issues that complicate collaborations between sectors, regions, and governments in the management of water resources, and conflicts of interest between departments can hinder cooperation and reduce the effectiveness of water resources and water environment management [3–6]. Integrated water resources management (IWRM), largely developed during the 1990s, is the most common organizational approach used in river basin management. IRWM is a process which promotes the coordinated development and management of water, land, and related resources in order to maximize economic and social welfare without compromising the sustainability of vital ecosystems [5,6]. The Chinese government has utilized the IWRM approach since 1988, when its first Water Law was enacted. However, the applications of IWRM in China proved ineffective and insufficient, as did similar applications in other parts of Asia and in Latin America [7,8]. Based on a recent investigation by Wang and Chen, the inefficiency of the IWRM and the difficulty implementing it in China may be ascribed to factors including amorphous definition, operational difficulty, departmental conflicts, lack of authority in river basin management, and the political moral

hazard that occurs when local governments are no longer solely responsible [4]. Struggling under a severe water crisis and the failure of IWRM, the Chinese government had to reform the water management system using a top-down design.

Early in the summer of 2007, a massive blue algae outbreak occurred in Taihu Lake, severely compromising the primary drinking water source of lakeside city Wuxi, in the Jiangsu Province. Motivated by the challenges posed by the water crisis, the local government of Wuxi nominated the head officials in charge of the Chinese Communist Party (CCP) and government to be river chiefs responsible for 64 major rivers. Remarkable improvements in water quality control were achieved in a short period of time, with the percent of major rivers that met water quality standards increasing from 53.2% to 71.1% in one year. This success highlighted the superiority of this newly established system, bringing it to the attention of other areas within the province [9]. Jiangsu province began nominating head officials as river chiefs throughout the whole province in September 2012. Over the following years, other provinces including Zhejiang, Anhui, Tianjin, etc., started to appoint CCP or government heads as river chiefs within their jurisdictions. Taking after the water pollution control plan, river management was achieved by establishing goals, breaking them down into manageable pieces, and delegating those among the governmental hierarchy (Figure 1).

At the end of 2016, the General Office of the CCP Central Committee and the State Council released the document 'The Opinions on Full Implementation of the River Chief System (RCS) Across the Country' [10], which provided plans that would protect water resources, control pollution, improve the environment, and restore natural communities nationwide. The establishment of the document signified that the RCS had rapidly transitioned from an emergent policy to a nationwide action. The RCS is fundamentally rooted in China's top-down administrative system and is characterized by the hierarchical systems of the party and the state [11]. By resolving issues created by incomplete legal systems and insufficient judicial guarantees, this innovative system may be the answer to current, complicated water issues [12]. The main tasks of the chiefs include water resources protection, shoreline management, water pollution prevention and control, water environment management, restoration of water ecology, and law enforcement. As an emerging system in response to complicated water issues, the deployment of the RCS has, understandably, encountered problems and difficulties. The non-statutory responsibilities of river chiefs, the over reliance on administrative power, the trans-provincial management of rivers and lakes, as well as the lack of public participation and social supervision all may impede the effective implementation of the RCS. These potential impediments will be discussed in the following sections of the manuscript. Nonetheless, in recent years, the RCS has proven itself as an effective method for solving complex collaborative problems in water management, which is deeply rooted in the country's unique political systems [4].

Figure 1 shows the organizational structure of a typical river chief system. The Office of the River Chief System (ORCS) is composed of chiefs and key staff from all relevant departments, all of whom offer assistance and support to river chiefs at different levels. The ORCS draws up water management plans and supervises and assesses river chiefs at lower levels. To promote horizontal cooperation between different units, a conference system has also been created. The conference system allows river chiefs and departmental leaders at different levels, and from different regions, to consult on cross-jurisdictional and large-scale issues [4]. The ORCS effectively coordinates the departments involved in river affairs and deals with the river issues including environmental protection, water distribution, wastewater management, respect of environmental policies or laws, land and resource utilization, organization between stakeholders, etc., as shown in Figure 1.

The rapid development of the RCS has garnered different views in the academic field, with studies covering a wide range of related topics, including recent developments, system construction, framework of implementation, management effects, environmental protection, etc. For example, Liu et al. summarized the development and implementation of the RCS using Foshan, in the Pearl River Delta region, as a case study to show that the RCS strengthened cooperation between administrative departments and established a complex yet effective water environment management structure [13].

Huang and Xu, on the other hand, presented a negative view on the implementation of RCS as they postulated that factors like power distribution, hierarchical systems, and public involvement could not be appropriately matched and would lead to the failure of environmental governance. They argued that, as a relatively closed water environment management system, the RCS was established primarily to strengthen the government's water management rather than to create a platform for diversified governance [14]. Ren suggested that this governance structure, created for rivers and lakes, has the features of a typical cross-administrative region, and that this type of additional cross-departmental collaborative governance is highly needed [15]. Wang and Chen analyzed the RCS by establishing an analytical framework according to collaborative governance theory, and concluded that the RCS was effective in tackling collaborative issues in water management, but they also noted its long-term impacts and sustainability cannot be accurately predicted [4].

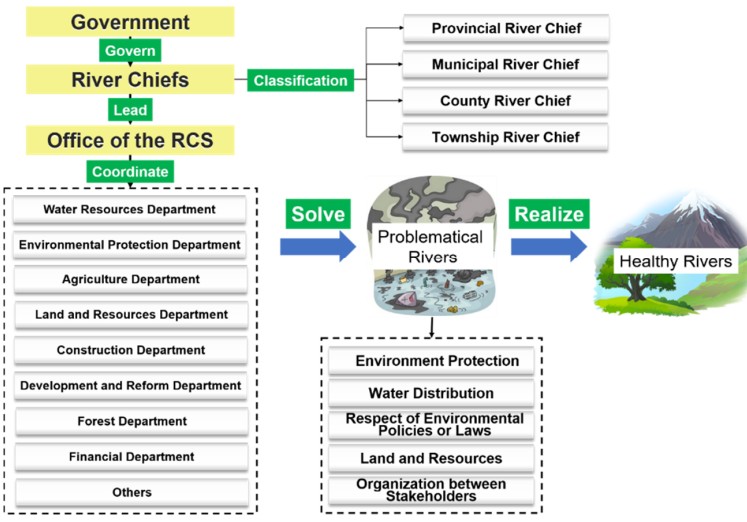

**Figure 1.** Schematic diagram of the river chief system (RCS).

Since the release of 'The Opinions on Full Implementation of the River Chief System Across the Country' in December 2016, as mentioned above, the RCS has been undergoing development nationwide. The main concerns of the current manuscript are as follows: (1) What are the recent developments under the institutional framework of the RCS? (2) What is the effectiveness of the RCS in preventing water pollution? (3) What are the weaknesses of the RCS and how might we facilitate its improvement? To address these questions, we have summarized the latest developments in the system, in technical strategies of implementation, in the legislation of the RCS, and in the public outreach. The changes in the water quality of important watersheds in China over the past several years were also evaluated. By describing current developmental progress and outstanding weaknesses and problems of the RCS, this paper proposes some practical approaches by which the innovative development of the RCS can continue.

## 2. Materials and Methods

Hohai University is a top academic institution in China that features multiple majors on water conservancy engineering and water environments. On April 28, 2017, the 'Research and Training Center for RCS in Hohai University' was founded with support from the Ministry of Water Resources. Being the first academic center entirely dedicated to RCS research across the nation, its founding effectively takes full advantage of Hohai University's theoretical knowledge and talent training, promoting theoretical and practical research on the RCS. Note that, at the time of writing the current manuscript, the authors were employed by the research center and committed to the study of the RCS. We conducted our survey through literature review, data collection, and case studies. Water quality data have been collected from the national bulletins and weekly reports from the Ministry of Water

Resources and the China National Environmental Monitoring Centre [16,17]. The literature review included both domestic and foreign studies on the implementation of the RCS in China. In addition to the scientific literature, we referred to announcements on the official websites of the National Ministry of Water Resources and the National Ministry of Ecology and Environment, which collect data from public resources including the 'National Water Resources Bulletin' and 'Health Status Report of Taihu Lake'. The legislation and implementation of laws and regulations on the RCS over the past three years were also analyzed and a summary was presented. We have been participating in the planning stages for the implementation of 'One River (or Lake), One Document' and 'One River (or Lake), One Strategy' for a number of rivers and we have held several discussions with relevant scholars, government officials, and enterprise representatives. Through these means, we have summarized the developments, strategies, outcomes, and weaknesses that have occurred during the past three years, following the full implementation of the RCS. Possible solutions to solve the problems have been proposed and discussed.

## 3. Results

### 3.1. Development of RCS after full Implementation as of December, 2016

After the release of 'the Opinions' in 2016, the Chinese government has been increasing its efforts to promote the development of the RCS. By the end of 2017, the Ministry of Water Resources and the Ministry of Environmental Protection had carried out a mid-term assessment on the progress implementation of the RCS. On November 20, 2017, the General Office of the CCP Central Committee and the State Council released 'Guiding Opinions on the Implementation of the Lake Chief System (LCS) in Lakes', and a general plan was made for the comprehensive implementation of both the RCS and LCS. On October 9, 2018, the Ministry of Water Resources issued 'The Opinions on the Promotion of the System from 'In Name' to 'In Practice' [18]. By the end of 2018, the RCS had been fully implemented in 31 provinces (including autonomous regions and municipalities) across the nation, with more than 300 thousand river chiefs nominated at the province, city, county, and township levels [19]. Furthermore, in 29 of the provinces, the system had been extended to the village level, with a total of 780 thousand village chiefs being appointed. Based on the practical experience by local governments concerning the unique environmental, hydrological, and social conditions, a river with a basin area larger than 10 km$^2$ is suggested to be included in the scope of RCS network. While smaller rivers or ditches with basin areas less than 10 km$^2$ are to be merged into the management of the upper level rivers. On March 5, 2020, the Ministry of Water Resources released the document 'Key Points of River and Lake Management Issues in 2020', which emphasized the realized and potential advantages of accelerating and promoting the system [20].

Here, we have summarized the developments since 2016 in the RCS from the perspectives of function expansion, implementation strategy, legislation process, and public participation. For the discussion, our findings on these developments have been categorized into the following sections: (1) innovative implementation of the lake chief system (LCS), (2) the standardized survey and evaluation of water environments: One River (or Lake), One Document, (3) the specific programming guide for RCS and LCS: One River (or Lake), One Strategy, (4) integration of the system into national and local laws and regulations, and (5) strengthening and implementation of the system for public participation.

### 3.1.1. The Implementation of the Lake Chief System (LCS)

Lakes are important parts of river systems and play invaluable roles in flood control, water supply, shipping, and ecology. China has 2693 natural lakes with areas greater than 1 km$^2$, making up a total lake area of 81,414.6 km$^2$, covering approximately 0.9% of the total land area [21]. Compared to rivers, the management of lakes has proven to be more difficult. Lakes have numerous problems including lake shrinking, disordered resource exploitation, water pollution, ecological degradation, and defective management systems, all of which hinder the effective management and control of lakes.

Aiming at solving these problems, the General Office of the CCP Central Committee and the State Council released the document 'Guiding Opinions on the Implementation of Lake Chief System (LCS) in Lakes' on November 20, 2017 [22].

Generally, the LCS network covers the management of lakes with areas larger than 1 km$^2$. The LCS stipulates that the lake chief at the highest level takes on the responsibility of lake management and protection [23]. That lake chief is therefore required to establish management and protection targets and goals, and enact the 'One Lake, One Strategy'. Lake chiefs are responsible for completing six main tasks, these include: (1) establishing strict lake space control, (2) protection of lakeshores, (3) protection of water resources including water pollution control, (4) comprehensive improvement of the lake water environment, (5) overseeing the ecological governance of lakes, and (6) perfecting law enforcement and supervision mechanisms [23]. The lake chief system was enacted nationwide by the end of 2018. A total of 24,000 hierarchical lake chiefs have been nominated at the provincial, city, county, and township levels for approximately 14,000 lakes across the nation. The implementation of the lake chief system effectively intensifies the management and protection of lakes, and initial results have been positive [24].

### 3.1.2. One River (or Lake), One Document: The Survey and Evaluation of Water Environment

The degree of pollution, pollution sources, ecological impacts, and vulnerability of rivers and lakes is different in different regions, different river basins, and different natural environments, all of which have different ages, traffic, and usage. It is essential to systematically characterize the baseline situations of rivers and lakes so that specific management and protection schemes can be put forward with specific measures that are logical and easily implemented based on the specific situation of any given river or lake. The Ministry of Water Resources published the documents 'The Guide for 'One River (or Lake), One Strategy' Establishment' and 'The Guide for 'One River (or Lake), One Document' Programming' in September 2017 and April 2018, respectively [25]. The aim of the two documents was to standardize and guide the development of any required technical work during the implementation of the RCS and LCS. The formulation of these documents manifested in remarkable progress in terms of system norms and organizational forms during the implementation of the RCS.

'One River (or Lake), One Document' is a collection of information regarding the natural characteristics, development, management protection, and dynamic changes of rivers and lakes. The Ministry of Water Resources has issued 'The Guide for 'One River (or Lake), One Document' Programming' in April 2018 [25]. A document for any specific river or lake should contain information on the water intake, sewage discharge, water quality, water ecology, coastline development, river channel utilization, and water-related projects and facilities.

Figure 2 is a representative schematic diagram of "One River (or Lake), One Document", in which the five informational sections include: river function information, basic channel information, management information, water resources information, and water environment and ecology information. Obtaining information pertaining to river function, channel properties, management and water resources is both feasible and relatively easy. Information regarding the water environment and ecology, on the other hand, must be broken down and evaluated in three component parts: the water quality status, water ecology status, and the determination of pollution sources. Water quality parameters, such as chemical oxygen demand (COD), total nitrogen (TN), ammonia nitrogen (NH$_3$-N), dissolved oxygen (DO), total phosphorus (TP), heavy metals, pH, turbidity, suspended solids, temperature, etc., have long served as the dominant factors used for evaluating a river's status. Owing to the complicated hydrological conditions and lack of universal mathematical models for the evaluation of water quality, specialized evaluation approaches are usually required to evaluate the monitored water quality data from a specific area and determine the water quality categories and the spatial and temporal variations of water quality. These parameters are used in concert with methods such as the single-factor assessment method, water quality grading method, the Nemerow pollution index, comprehensive pollution index, principle component analysis, and the fuzzy

comprehensive evaluation [26,27]. Nevertheless, these parameters only partially reflect the quality of the water body, which do not fully characterize the dynamic changes in water environments and water ecology. Consequently, the 'One River (or Lake), One Document' suggests that in addition to the conventional water quality parameters, evaluations of the ecological health of rivers are also required. For example, the parameters of the index of biotic integrity, or microbial index of biotic integrity are worth utilizing to provide both technical and theoretical support for the accurate assessment of river health [27–31].

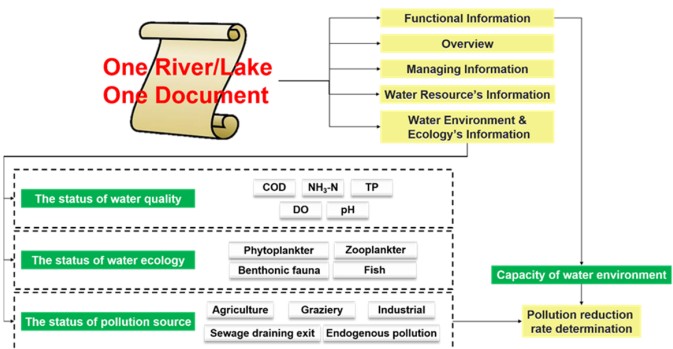

**Figure 2.** Schematic diagram of "One River (or Lake), One Document" [25].

### 3.1.3. One River (or Lake), One Strategy: The Programming Guide

The goal of the 'One River (or Lake), One Strategy' was to formulate specific River/Lake schemes based on the unique situations of each river or lake. Based on 'The Guide for 'One River (or Lake), One Strategy' Establishment', released by the Ministry of Water Resources, the programming of the 'One River (or Lake), One Strategy' should focus on the key issues facing river channels, and determine the primary management goals and the main tasks required for the protection of that river/lake. By comprehensively analyzing information about a specific river/lake, five lists should be created, these include the issue list, the goal list, the task list, the schedule list, and the responsibility list. The lists serve as concentrated check lists, derived from the information gathering stage, and as such, present the keys to objectively meeting the requirements for the successful management and protection of rivers and lakes. After the programming of 'One River (or Lake), One Strategy', specific measures are proposed for the prevention and control of water pollution, comprehensive treatment of the water environment, restoration of water ecology, and supervision of law enforcement. The detailed programming process of 'One River (or Lake), One Strategy' is shown in Figure 3.

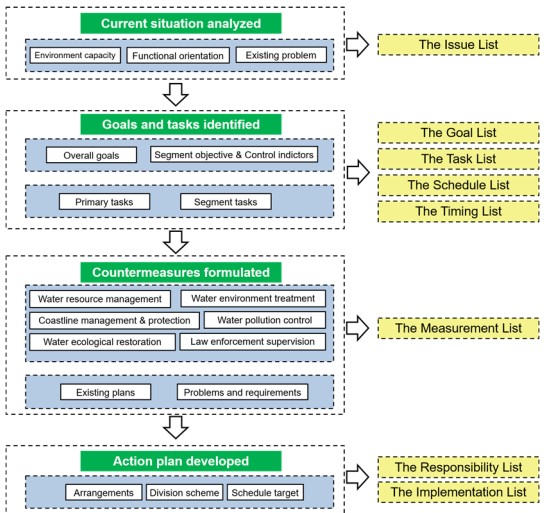

**Figure 3.** Programming processes of 'One River (or Lake), One Strategy' [25].

3.1.4. Legislation on RCS and LCS after Full Implementation

Despite the fact that the RCS has been experiencing dramatic progress in recent years, doubts regarding the RCS and LCS persist. A common criticism is that the system largely depends on administrative power and lacks the necessary legal strength, which means it will not last long-term. Therefore, the legislation of the RCS is an important and urgent issue to ensure its long-term effectiveness. On June 27, 2017, the RCS was formally written into the 'Law of the People's Republic of China on Prevention and Control of Water Pollution'. The fifth clause in the constitution now states:

RCS is to be implemented at province, city, township and village levels. The chiefs at different administrative levels are aimed to guide the tasks including water resource protection, water shoreline management, water pollution control and water environment remediation in their respective administrative areas.

This clause represents the highest legal regulation regarding the RCS in the Chinese legal system, and it formally supports this innovative system for river and lake management by empowering it with national legal support.

Subsequently, several local governments have formulated regulations and normative documents to provide specific details on the promotion and implementation of the local RCS from a legal perspective. There are two ways to legislate for the RCS, the first being special legislation to implement the system. 'The Regulations for RCS in Zhejiang Province' was the first such special legislation for the RCS in the nation [32], it had 19 clauses which provided comprehensive and systematic provisions on the connotation, scope of applications, responsibilities of the working organization, rewards, and punishments of the RCS. By the end of 2019, four additional provinces, including Zhejiang, Hainan, Jiangxi, and Liaoning, had adopted special regulations for their local RCS (Figures 4 and 5). In some other provinces, mostly near the east border of China as shown in Figure 4, no specific legislation on the RCS has been established. Instead, these provinces have attached collateral provisions which associate the RCS with existing local laws or regulations targeting river/lake management, water pollution control, or water resource management. For example, there are four clauses associated with the RCS in 'The River Management Regulations in Jiangsu Province', providing legal provisions for the system, defining the distribution of responsibilities, and the examination and the evaluation of the RCS.

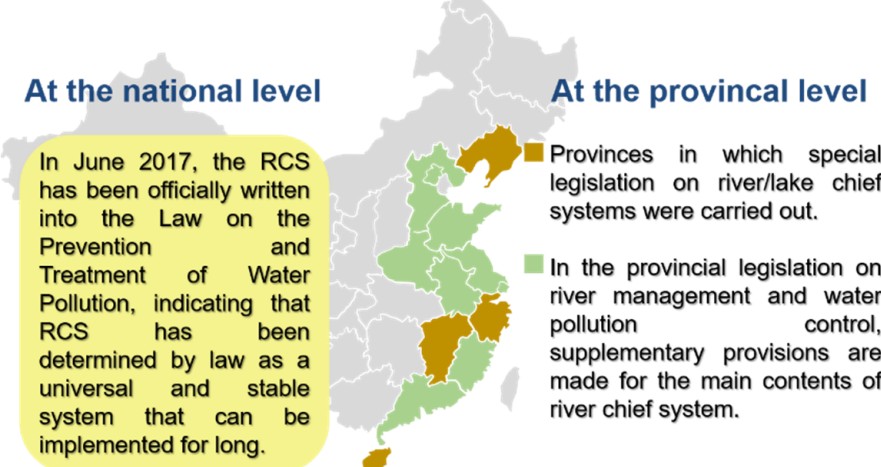

**Figure 4.** The legislation process of river/lake chief systems at national and provincial levels since 2016. (Source: Official websites of each provincial government).

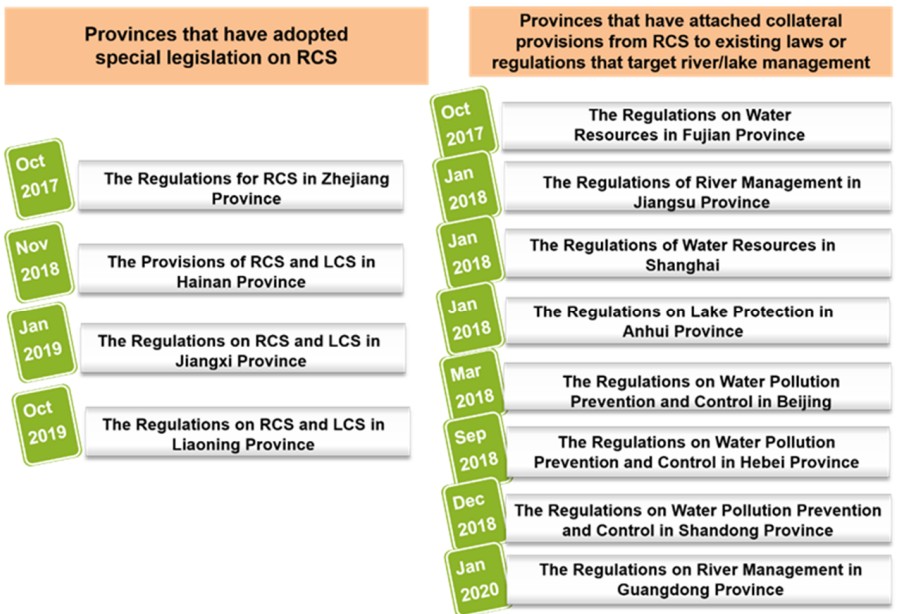

**Figure 5.** The calendar of RCS legislative decrees since 2017. (Source: Official websites of each provincial government).

### 3.1.5. Recent Progress in Public Participation in the Fully Implemented RCS

China's long-term government-led water management system has not encouraged public participation [33]. The RCS could address this deficiency by encouraging the participation of civil society [4]. To encourage collaboration between more sectors of society, river chief positions are not exclusive to government officers, and can also be filled by members of the public. The public has been encouraged to participate in the process of RCS planning and implementation, such as decision making, management, protection, supervision, and propaganda. In recent years, several new roles have emerged to encourage public participation even further; these include expert river chief, civilian river chief, dual river chief, entrepreneur river chief, etc.

In November 2019, the local government of Gulou District in Nanjing, China held a signing ceremony with Hohai University and hired 24 professors and vice professors to serve as 'expert river chiefs'. The experts were encouraged to actively participate in the routine river patrols, as well as the programming and implementation of 'One River (or Lake), One Strategy' to take advantage of their professional expertise [34]. In other cities in Sichuan and Anhui, the local governments also hired senior technicians or professors in water resources and water protection as expert river chiefs to provide guidance in the management of rivers and lakes. Civilian river chiefs consist of citizens or voluntary organizations that have been recommended by local governments to provide public supervision and oversight of the official river chiefs. For example, Hebei province has fully implemented the involvement of civilian river chiefs who carry out various forms of volunteer work, including river patrols and protection activities [35]. 'Dual river chief' refers to the mode in which the official river chief collaborates with the civilian river chief, and has gradually received more emphasis as the RCS has been implemented. The role of 'entrepreneur river chief' first appeared in Wuxi, the birthplace of the RCS. The entrepreneurs are engaged in the management and inspection of rivers. Familiar with the discharge conditions of pollutants, the entrepreneur river chiefs have experience that benefits the supervision and management of rivers in industrial areas. Under the discerning eyes of entrepreneur river chiefs, a strong restraining effect has been placed on polluting enterprises [36]. As of July 2018, more than 760,000 social river chiefs (including river cruisers and 208 river guards) at the province, city, county, and township levels had been nominated by local governments [25]. These efforts have improved the RCS management system by integrating public participation.

*3.2. The Outcomes of the Development of the RCS*

3.2.1. Water Quality Improvement in the Past Three Years

According to the 'National Water Resources Bulletin 2016' released on 11 July, 2016, the water quality in monitored rivers with grades worse than Grade V made up 9.8% of the total 235 thousand km$^2$ of rivers nationwide. The main pollutants were $NH_3$-N, TP, and COD. Among the 118 lakes, those with water qualities worse than Grade V made up 17.8% of the total lakes and eutrophicated lakes accounted for 78.6% of the total lakes, with the main pollutants being TP, COD, and $NH_3$-N.

The RCS is a representative local environmental governance policy that was fully implemented by the end of 2018. At this time, it is too soon to report outcomes or achievements of the RCS, which makes it difficult to form an accurate judgment on the real-world governance impact of the RCS. We have collected data from 'The weekly report on automatic water quality monitoring of major river basins and key sections of lakes and reservoirs in China' from 2016 to 2019 [16,17]. The monitoring stations covered 1698 sections from 978 major rivers across the nation. Data from 242 sampling points in 112 key lakes, (173 sampling points in 60 lakes and 69 sampling points in 52 reservoirs) were also included. According to the 'Annual Surface Water and Air Quality in China in 2019' released by the Ministry of Ecology and Environment, the monitored sections with water quality worse than Grade V accounted for 3.4% of the 1940 national surface water assessment sections, with COD and P representing the main pollutants [37]. The number of lakes with water quality worse than Grade V accounted for 7.3% of the 110 key lakes/reservoirs monitored, and the key pollutants were P and COD.

Figure 6 illustrates the annual data on the number of rivers or lakes that attained each water quality standard from 2015 to 2019. The number of rivers with water quality worse than Grade V has gradually decreased since 2015, as shown in Figure 6(a). In comparison, the number of rivers meeting Grade I to Grade II standards has increased. Figure 6(b) indicates that the number of key lakes meeting Grade I and Grade II standards also increased in the past years, but no clear changes were observed in heavily-polluted lakes. These results may indicate that the implementation of the RCS has increased the proportion of water bodies that have high water quality, i.e., Grade II or better. However, the RCS does not appear as effective in eliminating pollution from heavily polluted lakes. The long-term effects of the RCS on river and lake health are still unknown and need to be monitored further.

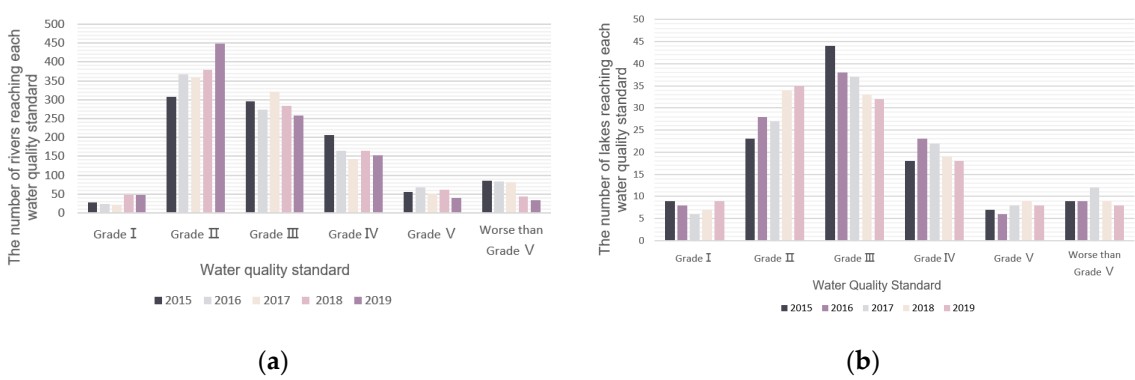

(**a**)                                                                                      (**b**)

**Figure 6.** Water quality of (**a**) 978 of the major rivers including the seven major river systems, and (**b**) 112 major lakes, including Taihu Lake, Dian Lake, etc., in China. Data source: Ministry of Ecology and Environment of the People's Republic of China.

3.2.2. Water Quality Improvement in the Yangtze River Basin

The Yangtze River basin covers 1.8 million square kilometers and 19 provinces, occupying 18.8% of the total land area of China [38]. Based on 'The weekly report on automatic water quality monitoring of major river basins and key sections of lakes and reservoirs in China' from 2016 to 2019 [16,17], we found that the overall water quality was good in the Yangtze River basin. Among the 510 sections monitored by the China National Environmental Monitoring Centre, the proportion that had water

quality of Grade III or better increased from 84.1% in 2016 to 91.7% in 2019, shown in Figure 7. At the same time, the proportion of sections with qualities worse than Grade V declined from 2.9% to 0.6%, clearly indicating that water quality has improved since the full implementation of the RCS. The watersheds around the Three Gorges Dam can be used as a representative, ecologically sensitive area in the Yangtze River basin, and the water quality in these watersheds also steadily increased. In 2016, the water quality of all monitored sections in this area had met Grade III standards, and by 2020, all monitored sections had reached or surpassed the Grade II standard.

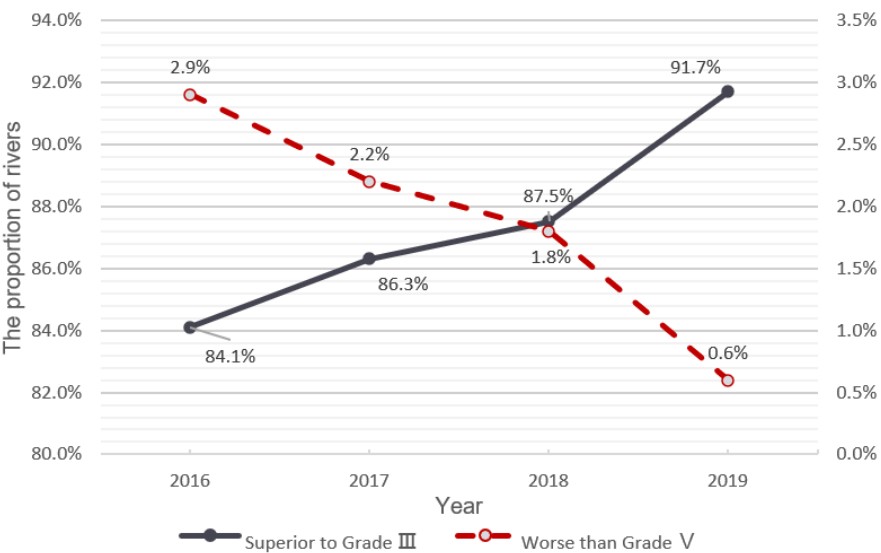

**Figure 7.** Water quality in the Yangtze River basin [16,17].

### 3.2.3. Water Quality Improvement in the Taihu Lake Basin

Taihu Lake, located in the center of the Yangtze River delta region, has an overall area of 2427.8 km$^2$. The lake has 228 tributary rivers, among which 170 and 58 rivers belong to the administrations of Jiangsu Province and Zhejiang Province, respectively. As mentioned above, the RCS policy was first proposed by the local government of the lakeshore city, Wuxi, as a response to blue algae blooms in Taihu Lake in 2007. The system has had a striking success in terms of water quality improvement in the Taihu Lake Basin.

The data in this section were obtained from the 'The Health Stats Report of Taihu Lake (2018)'. Figure 8 shows that, among the 108 key water function areas (42 in Jiangsu Province, 15 in Zhejiang Province, 6 in Shanghai, and 45 located on boundaries between provinces), 66.7% of the zones have met the water quality standards. A water function zone refers to an area that has a specific function, and has met the requirements of rational development, utilization, and protection of water resources in accordance with the regional water resources development and social needs. Figure 9 shows that the proportion of the 34 monitored sections with water quality better of Grade III or better has steadily increased. In this section, we have analyzed the water quality data of representative rivers and lakes to test the effects of the implementation of the RCS by local governments in China. The effectiveness in controlling water pollution at the local level was also evaluated.

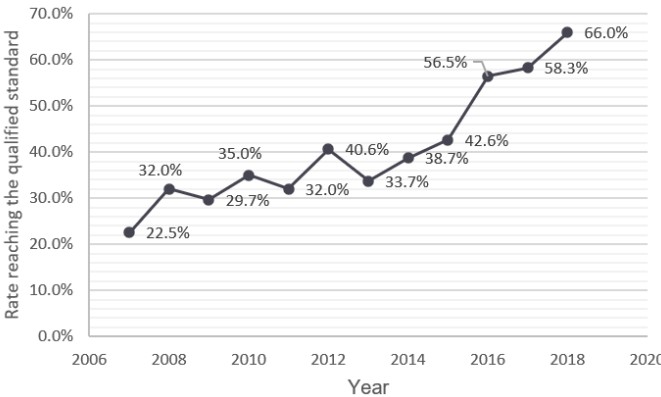

**Figure 8.** Change in the rate reaching the qualified standard among 108 key water functional areas in the Taihu basin from 2007 to 2018.

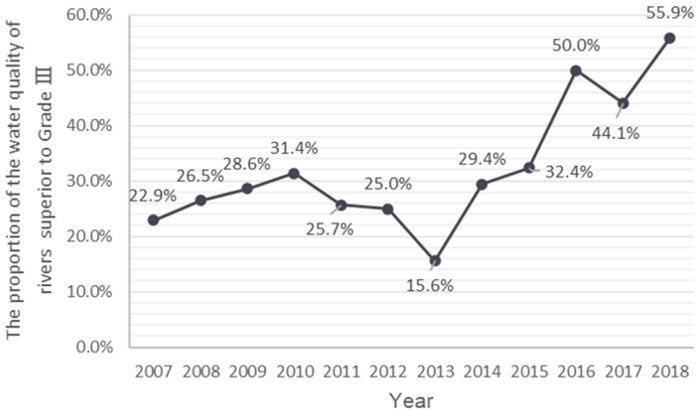

**Figure 9.** The proportions of upstream rivers with water quality superior to Grade III from 2007 to 2018 in Taihu Basin.

## 4. Discussion

### 4.1. The Superiority and Necessity of the RCS

As a response to the serious water environment issues by a public governance system, the RCS offered a significantly superior approach that has since become an indispensable tool in Chinese watershed management. River chiefs at all administrative levels are responsible for the supervision and management of the rivers and lakes, including protection of water resources, coastline management, water pollution control, water environmental management, leading the organization of the occupation of rivers, participating in the reclamation of lakes, and monitoring of excessive sewage, illegal sand mining, destruction of waterways, illegal fish hunting, etc. [6,9]. From the perspective of the top-down hierarchy, the river and lake chiefs are imbued with a strong sense of regional cooperation, which is conducive to increasing the efficiency of water resource management and ecological protection [4]. Under the guidance of a water pollution control plan, effective river/lake management will be achieved by breaking up the goals into manageable pieces and hierarchically delegating the targets. Moreover, strict evaluation and assessment will be integrated to track the chiefs and ensure they are efficient in achieving their target goals. Tackling the existing problems with Chinese watershed management, the RCS system sorted out the collaborative problems in water management among departments, local government, and various political levels. The RCS has also been proven effective in increasing social welfare in watershed management. In a recent study analyzing the economic and social welfare brought by the RCS, Liu et al. confirmed that one inevitable outcome of the RCS was the cross-regional Sustainable Water Resource Management Affairs (SWRMA) negotiations. The authors indicated that the system could help in avoiding transactional and external costs in

cross-regional SWRMA negotiations and that the efficiency of the RCS could be further enhanced by environmental negotiations and coordination mechanisms between different governments [39].

The superiority of the RCS largely hinges on the fact that it clarifies the previously confusing structure of watershed management. Previously, there was a lack of coordination and overlapping responsibilities between departments, which resulted in management loopholes in regards to environment protection, water conservancy, and land resources. As mentioned above, the failure of IWRM in China can be attributed to the problematic delegation of responsibilities between regional water administrative departments and basin-scale water management departments [4]. By appointing Party or government heads as river chiefs, multiple sectors were more easily coordinated and integrated under the organizational framework of the RCS. The action organically combined the tasks of channel improvements, flood control, water pollution control, environmental protection, ecological remediation, water allocation, and water conservancy. In this way, a water management approach with clear basis in traditional Chinese top-down hierarchy can be achieved. In a case study performed by Liu et al. where they surveyed the outcomes of the RCS in Foshan, China, the increased rate of rivers meeting water quality goals after implementation of the RCS clearly demonstrated the validity of the collaborative system in river pollution control, water conservancy, and ecological restoration [13].

### 4.2. The Weaknesses of RCS and the Possible Solutions

Since the full implementation of the RCS in 2016, the management system and working hierarchy have been substantially improved. However, there have been a variety of problems that this system has encountered, with numerous problems arising early during its implementation. These problems stem from three major sources. The first being that, in many cases, the main duties and responsibilities of river chiefs remain unclear. As mentioned above, while the amount of legislation on the river/lake chief systems has accelerated in the past three years, and in light of the release 'Law of the People's Republic of China on Prevention and Control of Water Pollution' and the numerous local regulations and normative documents, legal provisions on the statutory duties of river chiefs remain limited. The duties of river/lake chiefs, and other supervisory departments, have not been well defined and delegated, and the 'rule by man' approach can be seen in the real-world applications. Critics also argue that the overreliance on the personal power of river chiefs can cause severe responsibility gaps in the implementation of the RCS [4]. In the critical review by Huang and Xu, the authors argue that the strengthening of local authority could enable local river chiefs to combat or eliminate state power [14]. The authors point out that the RCS presents characteristics of a bureaucratic environmental governance, which has previously become an obstacle to effective water governance [14]. Consequently, legislation on the RCS is essential to define clear legal responsibilities of river chiefs at different administrative levels. It is foreseeable that along with the development of the RCS, legislation for the RCS will be needed nationwide. At the national level, it has been suggested that the government establish special legislation for the RCS by legislation supplementary laws or decrees. More provisions regarding the scope of application, the delegation of responsibility, and the legal liability of river chiefs should be established. As for the detailed responsibilities of the local hierarchy of chiefs, these will need to be stipulated in local legislation.

Second, the accountability and supervisory systems remain incomplete. The RCS aims to solve the key issue of assigning responsibility for specific water affairs. Along with this, the need for specific evaluation modes and methods for accountability has been identified. During the implementation of the river/lake chief systems, problems such as unclear distributions of responsibilities, unclear authority, and poor interoperability persist. The practice of shirking duties and responsibilities might occur more frequently if there is no clear system of accountability. In practice, the impartiality of an accountability review process may not be guaranteed because environmental bureaus or RCSO, which are responsible for the implementation of the process, are lower on the hierarchy than the river chiefs at administrative levels [40]. The accountability of river chiefs is imposed within the party-state hierarchy, by upper-level government, and is therefore susceptible to rent-seeking and corruption. According to the field

investigations carried out by Wang and Chen, no river chiefs have ever been relieved of their roles for poor management under the 'One-Vote Veto System' [4]. The current accountability mechanisms lack legal foundation. Additionally, the supervision and management mechanisms of the RCS are not perfect. As a result, there is a proportion of river chiefs that exist in name only, having no assigned tasks or responsibilities.

Third, the management of rivers and lakes has not been integrated at a large scale, with no clear comprehensive plan or coordination. The goals of the RCS are typically characterized by being trans-basin and trans-regional. However, where the RCS has been implemented on cross-border rivers, the administrative management standards and requirements sometimes differ between upstream and downstream areas, sometimes even on different sides of the same river. The effects of the differences between management on the upper reaches and lower reaches are difficult to quantify, which consequently leads to difficulty in assigning responsibility and further complicates river management. Specifically, the implementation of the RCS often lacks unification and coordination across multiple regions. The management of rivers and lakes is often unilateral, and comprehensive governance at the regional level is still lacking. The complexity of transboundary water issues has puzzled local governments as was shown in an important article surveying the science and negotiation theory of resolving boundary-crossing water issues. The article by Islam and Susskind called for the negotiated choice of decomposition among different regions, which has become an essential component of understanding, governing, and managing complex water systems [41]. An additional concern of the widespread deployment of the RCS is the variable financial support from different governments. Compared to the more developed eastern border of the nation, it may be more difficult for less developed regions to provide sufficient support for effective water governance.

The river/lake chief system is still constantly exploring new approaches for improvement. To solve the current problems and facilitate the transition from 'in name' to 'in practice', the following strategies need to be adopted to ensure the long-term effectiveness of the system. First, the implementation of the RCS and LCS need to be legally strengthened with the legal responsibilities of river chiefs clearly defined. Local laws or regulations on the RCS need to be enacted to specify the tasks of river chiefs and the distribution of rights and liabilities among the four-level river chief system. The perfection of laws and regulations could also help solve problems stemming from legal deficiencies and a problematic accountability system. Secondly, the management system needs to be improved, with a focus on coordinated and integrated systems seeking to achieve comprehensive governance by referring to the successful experience of comprehensive watershed management in other countries [5,6,41]. Finally, the government should continue to broaden communication channels and establish a sound public participation mechanism for the river chief system. A multi-channel guarantee for the realization of the river/lake chief system needs to be established from the perspectives of law, policy, management, technology, informatization, and public participation. The final task will be to set up a long-term government-led river management mechanism based on clear rules with participation from various sources including the public, the media, and individuals.

## 5. Conclusions

Since the release of 'The Opinions on Full Implementation of the River Chief System Across the Country' in December 2016, the RCS has been elevated to the national will and has experienced rapid development. This paper has mapped out the outcomes and weaknesses of the RCS that have come to light since its comprehensive implementation. The lake chief system, derived from the RCS, has been promoted throughout the nation as a response to complex lake management issues. Programming guides including 'One River (or Lake), One Document' and 'One River (or Lake), One Strategy' have created a standardized implementation strategy at the technical level. Laws and regulations at both the national and local levels have been issued to provide legal support and ensure the long-term effectiveness of the RCS. Unlike the long-term government-led water management approach, the RCS has the potential to promote a new role for the public as supervisors of government

water management projects, and recently created novel modes of public participation have promoted collaboration between government and civil society. The collected data have shown that after the full implementation of the RCS, water quality nationwide has improved, especially in terms of the proportion of water bodies that meet high water quality standards and in eliminating the existence of heavily polluted rivers. Despite its initial success and future potential, the fact that the RCS has only been implemented nationwide for three years means its long-term effectiveness in aspects such as water pollution control, water resource distribution, and water ecological restoration remain uncertain and need to be continuously monitored.

Despite substantial progress having been achieved, governments still encounter problems and difficulties in implementing the upgraded version of the system. The problems stemming from the imperfect formulation of laws and regulations, incomplete accountability and supervisory systems, unclear responsibilities and authority of each department, and the lack of unified and coordinated action need to be improved in the future. The system can be regarded as an active response to complex water issues and is deeply influenced by the unique political context and history of each specific region. The operation of the RCS depends on the culture of China's unitary system of government and the politics of the party-state hierarchy. The RCS has undergone a migration to national governance and has expanded our knowledge regarding the diversity of choices in water management tools, providing insights that may prove useful to other developing countries seeking to establish a river management system.

**Author Contributions:** Conceptualization, Y.L. and L.W.; data curation, J.T.; writing—original draft preparation, L.W.; writing—review and editing, Y.L. All authors have read and agreed to the published version of the manuscript.

**Funding:** This research was funded by the National Social Science Fund of China, grant number 17BZX035 and the Fundamental Research Funds for the Central Universities, grant number B200202199.

**Conflicts of Interest:** The authors declare no conflict of interest.

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
