# Peer review of "Full Implementation of the River Chief System in China: Outcome and Weakness"

_sustainability, doi:10.3390/su12093754_

Round 1
Reviewer 1 Report
see attached the edited manuscript and details below:
Review for Sustainability, April 2020
River Chief System after full implementation: Development, Strategy, and Outcome
By Li Y, Tong J, Wang L
General comments:
This paper could be (and I think “should be”) an answer to the very good paper of Huang and Xu (2019) published in Sustainability, cited by the authors but not really used. It’s a pity, since it’s obvious that Li et al. would like to reverse the conclusions presented by Huang and Xu (2019). Why not ? It would be very interesting.
Indeed, the authors Li et al. in this reviewed paper deals with the concept of river basin management, and specifically about the recent system of River Chief System, put in action in China. It was in 2016. And so, this paper proposes to discuss where these Chinese system is today ? It’s for sure fundamentally a very good question, but unfortunately not well managed in the present manuscript.
The authors do not use enough the opportunity of the great importance of literature on the river basin management issue, a key problem not only for the hydrologists, but also for both politicians, territorial technicians and citizens from cities and rural towns. I engage the authors to look at the excellent works from IWMI on that issue, at least :
F Molle, P Wester, P Hirsch, JR Jensen, H Murray-Rust, V Paranjpye, S Pollard, P Van der Zaag, 2007. River basin development and management. In Molden et al. Ed. Sc. : Water for Food, water for Life, IWMI Books, Reports H040208, 585-624.
And many others: Islam and Susskind (2018) in J Hydrology; Molle F (2009) in Geoforum,…
The introduction didn’t highlight the key question that the paper will discuss. The literature is not well used. Then the part of results is not well organized, not enough specific. It is surely due to the weakness of the introduction which didn’t focus on the key scientific issue of the paper. And there were many confusions between results and discussion. The discussion is curiously without any citations. And moreover the examples used to prove the efficiency of RCS are not well presented or chosen. Consequently the conclusion and the abstract should be revisited.
Based on the so weaknesses aspects of the manuscript and due to the interest of the subject, I propose to reconsider the manuscript after major revision the authors will find below.
To conclude, I suggest to the authors to define better which is the scientific question and how to do it to response. The authors wanted to use concrete data to prove that RCS and LCS are good systems. Unfortunately these systems are in action only from 3 years and they didn't provide an exhaustive view of all the data of water quality got in China. It could be interesting but will request a huge work of data mining. And maybe, it would be another paper. But it would be already a good paper with just a rigorous presentation of the full implementation of RCS with a honest scientific discussion, without any need to require water quality data.
At last, I engage the authors to go through this recent paper:
Liu P, Pan Y, Zhang W, Ying L, Huang W, 2020. Achieve Sustainable development of rivers with water resource management - economic model of river chief system in China. STOTEN, 708, doi.org/10.1016/j.scitotenv.2019.134657
And to comment their conclusion:
“Based on these results, this study provided the following suggestions: the river chief and local authorities of RCS should be stakeholders; the major members of RCS should be the local authorities influenced mostly by SWRMA quantities; RCS should actively advertise the benefits and the cost-down of SWRMA.”
By this example, I wanted to show you the interest of “local actions”, that never you discussed. What about the individual people, the individual needs and requests, etc…
And the new title could be:
Full implementation of the River Chief System in China: strengths and weaknesses
Comments and recommendations:
Introduction
L31: Why to design Chinese ? It is not a specific issue from China but everywhere in the world. Then add some other global citations.
Molle F, 2009. River-basin planning and management: The social life of a concept. Geoforum, 40: 484-494.
Islam S, Susskind L, 2018. Using complexity science and negotiation theory to resolve boundary-crossing water issues. J. Hydrology, 562: 589-598.
L33-38: I am not sure this example feed well the introduction. It would be more attractive for an international publication to make here a short review to connect RCS with the IRBM (integrated river basin management) largely developed in the world from the 1990s.
L47: From 2016, it is only at maximum 4 years of RCS implementation ! You should modulate your text to clarify your idea. Maybe precise here than you talked only of Jiangsi experienced from 2012 ??
L52: Too much Chinese references. Maybe number 9 will be enough.
L60: On this scheme, it's not only the RCS organization but also a specific vision of the main goal about the environmental issues. Is it the reality ? or your reality ?
The RCS would be implemented only to control the river pollutions ? It is exactly what you should absolutely to discuss.
L63-64: No word about the environmental issues ? It is in opposition with the fig. 1 !??
L67: As on the fig 1, you focus here on the environment issues. But the cooperative management was about only a question of environment ? or it was also some issues on water distribution, wastewater management, respect of environmental policies or laws, organization between stakeholders, etc... It is what you must discuss.
L68: I don't agree. They said "should be able" to achieve environmental goals, but it does not work. "should be able" is absolutely not "were able".
You should come back to the very good paper of Huang and Xu (2019) to precise what is the fundamental goal of RCS by definition from the Chinese Law in 2016. And after, you should provide your questioning you would like to discuss in your paper.
L76-81: I am not sure it is the good place. I will put it in "materials and methods" (just before your line 92). The introduction should raise your questioning.
L83-84: should be related to line 44.
L89-90: Comments on your introduction
So, to achieve this, you should use your introduction to highlight all the positive and negative reactions provided by the RCS experience in China. Maybe you could use more the paper of Huang and Xu (2019) which is very negatively critical of this system. They said:
"RCS was established mainly to strengthen the government’s water
management rather than to create a platform for diversified governance" and concluded on: "which might become an obstacle to effective water governance."
Here, you have a marvelous disruptive statement you could discuss.
Materials and Methods
L92: I'll put here your lines 76-82
L92: precise what kind of data ?
L103: so in your line 47, it would be better to emphasize this: you comment the RCS experience in China from last 3 years. It's clear and additional work for the international community.
Results
L160: I didn't understand the legend ?
And with this figure, it seems that the RCS and LCS networks are not applied nationwide, could you explain ?
It is apply only on the East border of China ?
L165: so, what about Jiansu (cf line 39) ?
L166: maybe the figure 3 could listed these 19 clauses.
L168: So Fig 2 and 3 are only for 5 provinces ?? Then you should adapt their title in function. Which province was the first ? It’s not clear between line 39 with Jiangsu and line 165 with Zhejiang, I quote you : “The Regulations for RCS in Zhejiang Province’ was the first such special legislation” ??
L170: I didn't understand what you mean ? Explain
L176: again, I didn't understand the title of the figure with the legend text inside the figure. It seems to be more a calendar of RCS legislative decrees.
L177: so, it is not a result but your statement that could be in the discussion part ?
L178: you mean today it is not the case ?
L179: you mean arrangements, facilities ? by supplementary laws or decrees ?
L181: I got some difficulties with this paragraph. Did you express here your own perception ? Then it whould be in the discussion part.
Or did you express a common evaluation of RCS already published ? Then you should provide some citations.
L185: you mean citizens ?
L205: areas
L208: you mean nominated, so designed by who ? and are they paid ? or any other advantages ? something else ?
Comments: This part could be shortened with clear details on what were the first clauses at the beginning and then what were the addition. Your current text is too long and not enough organized. You should also provide the definition of a “river” and a “lake” for the RCS and LCS network: for example, which size ? There is a minimum amount of water provision ? Or is it related to the number of users ? or…????
L216-218: What is the link with what you already said in lines 109-112 with citation 16: "On November 20, 2017, the General Office of the CCP Central Committee and the State Council released the document of ‘Guiding Opinions on the Implementation of the Lake Chief System (LCS) in Lakes’, and a general plan was made for the comprehensive implementation of both the RCS and LCS. On October 9, 2018, the Ministry of Water Resources issued the document ‘The Opinions on the Promotion of the System from ‘In Name’ to ‘In Practice’ "
L219: so, it would be more comprehensive that you develop this part before your description of the complements done by some specific provinces, such as discussed from line 162 to 209.
L223: you mean one document for one water body ? so each considered river and each considered lake have each one one document ?
L237: the quality parameters should be listed exhaustively. Then what about heavy metals ? POPs ?
L239: who pay to get these expensive paramters to be measured ??
L242: explain ?
L244-250: In fact, it is not so simple to qualify the quality of the waters. It depends about the final use. So the few explanations provided here are not satisfying. Since it is not the core of this paper, it would be better to avoid this issue. Just you explain in few words that the RCS and LCS policy to characterize the water quality.
The papers 29 to 31 should/could be more exploited and described to legitimate the use of this index in your paper. But it's not this index you plot in figures 6 to 9 , why ?
L273: oh ! Are they the main pollutants ? What about the trace elements such as A, Cr, B (Xiao, Wang et al, 2019, STOTEN), metal pollution (Zhang, Qin et al, 2018, STOTEN) and microplastics (Wang, Ndungu, 2017, STOTEN), and so on...??
L286-288: I didn't understand: it is the opposite of what you say Line 275 ! And I agree with the statement of the line 275.
However if you really want to prove the efficiency of RCS&LCS on water qualiy, you should provide data. But I didn't think its the aim if this paper, then I would delete this paragraph (from line 268).
L289: oh, it should be reported inside the method paragraph.
L294: you cannot be so affirmative. Moreover in this result part, you should just describe te data, and discuss on them !
BTW, it would be good to present these 2 graphs with the STD values. And more than the % the numbers of water bodies would be better (number of rivers, numbers of lakes). You should described that in general there are no significant evolution and sometimes more degradation.
L295: I don't understand the two vertical legends of your 2 graphs. For b, you mean: "Number of lakes ..." and not rivers ?
Why you used "Env quality standard" for a, and "degree of contamination" for b ?
I would write "Env quality standard" for both.
Comments:It could be interesting to show these data. So you should describe how did you do to calculate the index of quality for rivers and lakes, to describe also how many measurements per station, etc…
L303: oh no ! you cannot compare interannual data and here intermonthly data. The evolution along the year could be explained by the climatic seasons, by the cycle of human activities, by the economic conjonctures, etc...
Then this paragraph should be deleted.
L328: explain this concept in your text
L331: meaning ? is it upstream rivers ?
Comments: If you want to describe some specific cases to demonstrate the impacts of the RCS&LCS, then you should describe the study cases before. Without that, it is difficult to be convinced.
Discussion
L345: but Huang and Xu, 2019, had the opposite discours.
L347: innovative for China maybe, but not with other integrated river management systems in the world.
L352: then it is not done yet, isn't it ?
So the RCS&LCS network is not entirely set up ?
L362: It would be very intersting to discuss that point. is it enough ? What about the task of water distribution ? water allocation ?
L373: is it a necessity ??? you should read some papers from the outside China.
L337: where ? how ? by who ? Your sentence is an assertion.
L385: YES
L391: always of nowadays ? or it was before?
If it was before, write : were.
L396: so it should be described in your result part.
L400: and not by being multi-actors ? It is here the key question of this paper.
L402: no, not "can" but "should". It is obliged to have many changes between upstream and downstream at each environmental organization level. Etc...
L403-405: YES
L407: oh ! I am surprised since I thought that RCS/LCS systems were to avoid that issue.
L412: Yes, it is exactly what your paper should discuss. How to do it ? your ideas ? the requirements ?
L413: explain
L419: and what about the individual needs and wishes ?
Conclusion
L423: not really
L424: yes, but should be better organized.
L425: why this sentence to emphasize the LCS ? It is not what we have inside the text.
L426: it is exactly what your text should develop. Maybe you could organize your paragraphs around this concern:
- RCS related to national, regional scales
- RCS related to local, individual requirements
L430: absolutely not ! You cannot write a so affirmative result.
L437-440: It is exactly what we would like to be demonstrated in the paper. But it is not done yet.
Resume
L17 : ou have listed the actions done to implement the RCS. But you didn't analyze the implementation actions. It would be very interesting to do it.
L23 : No, your paper is very far away to prove that assertion.
END------------------------------

Author Response
Response to Reviewer 1 Comments
Point 1: This paper could be (and I think “should be”) an answer to the very good paper of Huang and Xu (2019) published in Sustainability, cited by the authors but not really used. It’s a pity, since it’s obvious that Li et al. would like to reverse the conclusions presented by Huang and Xu (2019). Why not? It would be very interesting. Indeed, the authors Li et al. in this reviewed paper deals with the concept of river basin management, and specifically about the recent system of River Chief System, put in action in China. It was in 2016. And so, this paper proposes to discuss where the Chinese system is today? It’s for sure fundamentally a very good question, but unfortunately not well managed in the present manuscript.
Response 1: We appreciate the comments raised by the reviewer on the conception of the manuscript. The reviewer is right that we have reviewed paper deals with the concept of river basin management and the recent developments of RCS. The authors agree that the current manuscript is not well managed and organized, which needs to be much improved.
However, it is not our original purpose to reverse the conclusions presented by Huang and Xu (2019), who clearly presented a negative view on the implementation of RCS. They postulated that factors like power distribution, hierarchical systems, and public involvement could not be appropriately matched and would lead to the failure of environmental governance. We have merely summarized the recent developments and analyzed the strategies including ‘One River (or Lake), One Document’ and ‘One River (or Lake), One Strategy’ during the implementation of RCS/LCS. The current manuscript is more like a neutral summarization of recent developments and weaknesses of RCS, rather than a debate article reversing any other paper. We have described some current developmental progress and outstanding weaknesses and problems of the RCS. Some practical approaches by which the innovative development of the RCS can continue were also proposed. Accepting the reviewer’s valuable comment, we have re-arranged the current manuscript and the reviewer could clearly observe the efforts performed by the authors.
Point 2: The authors do not use enough the opportunity of the great importance of literature on the river basin management issue, a key problem not only for the hydrologists, but also for both politicians, territorial technicians and citizens from cities and rural towns. I engage the authors to look at the excellent works from IWMI on that issue, at least :
F Molle, P Wester, P Hirsch, JR Jensen, H Murray-Rust, V Paranjpye, S Pollard, P Van der Zaag, 2007. River basin development and management. In Molden et al. Ed. Sc. : Water for Food, water for Life, IWMI Books, Reports H040208, 585-624.
And many others: Islam and Susskind (2018) in J Hydrology; Molle F (2009) in Geoforum,…
Respond 2: Accepting the reviewer’s suggestions, the mentioned literature have been cited and the review on river basin management has been strengthened on Lines 34 to 46 and on Lines 476 to 478 in the revised version:
Integrated water resources management (IWRM), largely developed during the 1990s, is the most common organizational approach used in river basin management. IRWM is a process which promotes the coordinated development and management of water, land, and related resources in order to maximize the economic and social welfare without compromising the sustainability of vital ecosystems [5,6]. The Chinese government has utilized the IWRM approach since 1988, when its first Water Law was enacted. However, the applications of IWRM in China proved ineffective and insufficient, as did similar applications in other parts of Asia and in Latin America [7,8]. Based on a recent investigation by Wang & Chen, the inefficiency of the IWRM and the difficulty implementing it in China may be ascribed to factors including amorphous definition, operational difficulty, departmental conflicts, lack of authority in river basin management, and the political moral hazard that occurs when local governments are no longer solely responsible [4]. Struggling under a severe water crisis and the failure of IWRM, the Chinese government had to reform the water management system using a top down design.
The article by Islam & Susskind called for the negotiated choice of decomposition among different regions, which has become an essential component of understanding, governing, and managing complex water systems [41].
Point 3: The introduction didn’t highlight the key question that the paper will discuss. The literature is not well used. Then the part of results is not well organized, not enough specific. It is surely due to the weakness of the introduction which didn’t focus on the key scientific issue of the paper. And there were many confusions between results and discussion.
Respond 3: We appreciate the reviewer’s valuable comment. The current manuscript is more like a neutral summarization of recent developments, legislation process, implementation strategy and weaknesses of RCS, rather than a debate article reversing any other manuscript. The authors have re-arranged and revised the sections in results and discussion in the previous version of manuscript. We will be much appreciated if the reviewer could see the efforts of the authors in improving the overall quality of the manuscript.
Point 4: The discussion is curiously without any citations. And moreover the examples used to prove the efficiency of RCS are not well presented or chosen.
Respond 4: The discussion has been carefully revised by citing more literature and with more discussion. As mentioned in response to the above comment, the authors have no intention to prove the efficiency of RCS, but just summarize the recent developments and weaknesses of RCS. The examples used have been checked and modified as suggested.
Point 5: Consequently the conclusion and the abstract should be revisited.
Respond 5: The conclusions and the abstract have been revised as suggested.
Abstract: Despite having explored various modes of water management over the past three decades, the water crisis persists and the Chinese government has been required to revolutionize river management from the top down. The River Chief System (RCS), which evolved from small scale, local efforts to manage rivers starting in 2007, is an innovative system that coordinates between existing ‘fragmented’ river/lake management and pollution control systems to clearly define the responsibilities of all concerned departments. The system was promoted from an emergent policy to nationwide action in 2016, and ever since, has undergone steady development. We have analyzed recent developments in the system from the perspectives of functional expansion, implementation strategies, legislative processes, and public outreach after the full implementation of the RCS. By collecting data over the past several years, the changes in the water quality of representative watersheds in China were evaluated to assess the outcomes of RCS implementation. Finally, a summary of the weaknesses and outstanding problems of the system is presented, putting forward a multi-channel strategy for the long-term stability and effectiveness of river/lake chiefs, and promoting the RCS as a suitable solution to the collaborative and jurisdictional issues in water management in China.
- Conclusions
Since the release of ‘The Opinions on Full Implementation of the River Chief System Across the Country’ in December 2016, the RCS has been elevated to the national will and has experienced rapid development. This paper has mapped out the outcomes and weaknesses of the RCS that have come to light since its comprehensive implementation. The lake chief system, derived from the RCS, has been promoted throughout the nation as a response to complex lake management issues. Programming guides including ‘One River (or Lake), One Document’ and ’One River (or Lake), One Strategy’ have created a standardized implementation strategy at the technical level. Laws and regulations at both the national and local levels have been issued to provide legal support and ensure the long-term effectiveness of the RCS. Unlike the long-term government-led water management approach, the RCS has the potential to promote a new role for the public as supervisors of government water management projects, and recently created, novel modes of public participation have promoted collaboration between government and civil society. The collected data have shown that after the full implementation of the RCS, water quality nationwide has improved, especially terms of the proportion of water bodies that meet high water quality standards and in eliminating the existence of heavily-polluted rivers. Despite its initial success and future potential, the fact that the RCS has only been implemented nationwide for three years means its long-term effectiveness in aspects such as water pollution control, water resource distribution, and water ecological restoration remain uncertain and need to be continuously monitored.
Despite substantial progress having been achieved, governments still encounter problems and difficulties in implementing the upgraded version of the system. The problems stemming from the imperfect formulation of laws and regulations, incomplete accountability and supervisory systems, unclear responsibilities and authority of each department, and the lack of unified and coordinated action need to be improved in the future. The system can be regarded as an active response to complex water issues and is deeply influenced by the unique political context and history of each specific region. The operation of the RCS depends on the culture of China’s unitary system of government and the politics of the party-state hierarchy. The RCS has undergone a migration to national governance and has expanded our knowledge regarding the diversity of choices in water management tools, providing insights that may prove useful to other developing countries seeking to establish a river management system.
Point 6: Based on the so weaknesses aspects of the manuscript and due to the interest of the subject, I propose to reconsider the manuscript after major revision the authors will find below.
Respond 6: We appreciate the reviewer’s valuable comment and have re-arranged and revised the manuscript carefully. We will be much appreciated if the reviewer could see the efforts of the authors in improving the overall quality of the manuscript.
Point 7: To conclude, I suggest to the authors to define better which is the scientific question and how to do it to response. The authors wanted to use concrete data to prove that RCS and LCS are good systems. Unfortunately these systems are in action only from 3 years and they didn't provide an exhaustive view of all the data of water quality got in China. It could be interesting but will request a huge work of data mining. And maybe, it would be another paper. But it would be already a good paper with just a rigorous presentation of the full implementation of RCS with a honest scientific discussion, without any need to require water quality data.
Respond 7: We appreciate the reviewer’s valuable comment. The identification of the scientific question will definitely attract the attention of readers and aid in improving the quality of articles. Actually, as in response to the above comment, the authors have merely summarized the recent developments and analyzed the strategies including ‘One River (or Lake), One Document’ and ‘One River (or Lake), One Strategy’ during the implementation of RCS/LCS. The current manuscript is more like a neutral summarization of recent developments and weaknesses of RCS, rather than a debate article reversing any other manuscript. We will be much appreciated if the reviewer could see the efforts of the authors in improving the overall quality of the manuscript. Also, we have stressed the questions of this study in the section of Introduction on Lines 110 to 118 in the revised version:
The main concerns of the current manuscript are as follows: (1) What are the recent developments under the institutional framework of the RCS? (2) What is the effectiveness of the RCS in preventing water pollution? (3) What are the weaknesses of the RCS and how might we facilitate its improvement? To address to these questions, we have summarized the latest developments in the system, in technical strategies of implementation, in the legislation of the RCS, and in the public outreach. The changes in the water quality of important watersheds in China over the past several years were also evaluated. By describing current developmental progress and outstanding weaknesses and problems of the RCS, this paper proposes some practical approaches by which the innovative development of the RCS can continue.
Point 8: At last, I engage the authors to go through this recent paper:
Liu P, Pan Y, Zhang W, Ying L, Huang W, 2020. Achieve Sustainable development of rivers with water resource management - economic model of river chief system in China. STOTEN, 708, doi.org/10.1016/j.scitotenv.2019.134657
And to comment their conclusion:
“Based on these results, this study provided the following suggestions: the river chief and local authorities of RCS should be stakeholders; the major members of RCS should be the local authorities influenced mostly by SWRMA quantities; RCS should actively advertise the benefits and the cost-down of SWRMA.”
By this example, I wanted to show you the interest of “local actions”, that never you discussed. What about the individual people, the individual needs and requests, etc…
Respond 8: Accepting the reviewer’s suggestion, we have made comments to the article on Lines 405 to 411 in the revised version:
In a recent study analyzing the economic and social welfare brought by the RCS, Liu et al. confirmed that one inevitable outcome of the RCS was the cross-regional Sustainable Water Resource Management Affairs (SWRMA) negotiations. The authors indicated that the system could help in avoiding transactional and external costs in cross-regional SWRMA negotiations and that the efficiency of the RCS could be further enhanced by environmental negotiations and coordination mechanisms between different governments [39].
Point 9: And the new title could be: Full implementation of the River Chief System in China: strengths and weaknesses
Respond 9:We appreciate the reviewer’s valuable suggestion on the modification of the title of the manuscript. We believe that if the strengths and weaknesses of RCS are comprehensively discussed and compared, the conception and value of the manuscript will be much improved.
In the current version of manuscript, we have summarized the weaknesses of RCS, such as non-statutory responsibilities of river chiefs, the over reliance on administrative power, the trans-provincial management of rivers and lakes, as well as the lack of public participation and social supervision have also been discussed in our manuscript. We politely disagree with the reviewer that the expression ‘strengths’ is suitable in the caption, since the authors have just summarized the recent developments of RCS and LCS after the full implementation since 2016. The implementation of LCS, the legislation of RCS and the upgraded public participation could not be truly classified into strengths of the system. Besides, we have analyzed the strategies including ‘One River (or Lake), One Document’ and ‘One River (or Lake), One Strategy’ during the implementation of RCS/LCS, which could not be classified into ‘strengths’ either. By referring to the reviewer’s valuable suggestion and the purpose of our manuscript, we have modified the title to ‘Full implementation of the River Chief System in China: outcome and weakness’.
Comments and recommendations:
Point 10: Introduction
L31: Why to design Chinese ? It is not a specific issue from China but everywhere in the world. Then add some other global citations.
Molle F, 2009. River-basin planning and management: The social life of a concept. Geoforum, 40: 484-494.
Islam S, Susskind L, 2018. Using complexity science and negotiation theory to resolve boundary-crossing water issues. J. Hydrology, 562: 589-598.
Respond 10: The sentence has been revised without specifically stressing the Chinese situation and the aforementioned citations have been added on Lines 31 to 34:
There are challenging issues that complicate collaborations between sectors, regions, and governments in the management of water resources, and conflicts of interest between departments can hinder cooperation and reduce the effectiveness of water resources and water environment management [3-6].
Point 11: L33-38: I am not sure this example feed well the introduction. It would be more attractive for an international publication to make here a short review to connect RCS with the IRBM (integrated river basin management) largely developed in the world from the 1990s.
Respond 11: Accepting the reviewer’s suggestion, we have made a short review on the connection between RCS and IWRM as well as remaining the current example explaining the origin of RCS on Lines 34 to 46 in the revised version.
Integrated water resources management (IWRM), largely developed during the 1990s, is the most common organizational approach used in river basin management. IRWM is a process which promotes the coordinated development and management of water, land, and related resources in order to maximize the economic and social welfare without compromising the sustainability of vital ecosystems [5,6]. The Chinese government has utilized the IWRM approach since 1988, when its first Water Law was enacted. However, the applications of IWRM in China proved ineffective and insufficient, as did similar applications in other parts of Asia and in Latin America [7,8]. Based on a recent investigation by Wang & Chen, the inefficiency of the IWRM and the difficulty implementing it in China may be ascribed to factors including amorphous definition, operational difficulty, departmental conflicts, lack of authority in river basin management, and the political moral hazard that occurs when local governments are no longer solely responsible [4]. Struggling under a severe water crisis and the failure of IWRM, the Chinese government had to reform the water management system using a top down design.
Point 12: L47: From 2016, it is only at maximum 4 years of RCS implementation! You should modulate your text to clarify your idea. Maybe precise here than you talked only of Jiangsu experienced from 2012?
Respond 12: We appreciate the reviewer’s comment. We think the incorrect description of the implementation of RCS led to the confusion of the reviewer. Actually, the RCS has been fully implemented across the country since 2016, but the first attempt of RCS was generated since 2007 in Jiangsu Province. The system has been progressively implemented to other provinces since 2007 to 2016. At the end of 2016, the General Office of the CCP Central Committee and the State Council released ‘The Opinions on Full Implementation of the River Chief System Across the Country’, and in this manuscript we have focused on the updates of the system after the full implementation across the country.
The reviewer is right that Jiangsu firstly implemented the system since 2012 with other provinces following the experience since then. We have carefully checked the related expressions here and make expressions with more clarity on Lines 53 to 65 in the revised version:
This success highlighted the superiority of this newly established system, bringing it to the attention of other areas within the province [9]. Jiangsu province began nominating head officials as river chiefs throughout the whole province in September 2012. Over the following years, other provinces including Zhejiang, Anhui, Tianjin, etc. started to appoint CCP or government heads as river chiefs within their jurisdictions. Taking after the water pollution control plan, river management was achieved by establishing goals, breaking them down into manageable pieces, and delegating those among the governmental hierarchy (Figure 1).
At the end of 2016, the General Office of the CCP Central Committee and the State Council released the document ‘The Opinions on Full Implementation of the River Chief System (RCS) Across the Country’ [10], which provided plans that would protect water resources, control pollution, improve the environment, and restore natural communities nationwide. The establishment of the document signified that the RCS had rapidly transitioned from an emergent policy to a nationwide action.
Point 13: L52: Too much Chinese references. Maybe number 9 will be enough.
Respond 13: Modification has been made as suggested.
Point 14: L60: On this scheme, it's not only the RCS organization but also a specific vision of the main goal about the environmental issues. Is it the reality ? or your reality ?
The RCS would be implemented only to control the river pollutions ? It is exactly what you should absolutely to discuss.
Respond 14: We appreciate the reviewer’s valuable comment. We have carefully checked the aim of RCS and modifications have been made in the scheme:
Figure 1. Schematic diagram of the river chief system (RCS).
Point 15: L63-64: No word about the environmental issues? It is in opposition with the fig. 1 !??
Respond 15: We appreciate the reviewer’s valuable comment. The expressions on environmental issues have been added on Lines 91 to 93 in the revised version:
The rapid development of the RCS has garnered different views in the academic field, with studies covering a wide range of related topics, including recent developments, system construction, framework of implementation, management effects, environmental protection, etc.
Point 16: L67: As on the fig 1, you focus here on the environment issues. But the cooperative management was about only a question of environment? or it was also some issues on water distribution, wastewater management, respect of environmental policies or laws, organization between stakeholders, etc... It is what you must discuss.
Respond 16: Appreciating the reviewer’s valuable comment, the contents of Figure 1 has been revised and we have made more discussion on the aforementioned issues on Lines 84 to 87, in the revised version:
The ORCS effectively coordinates the departments involved in river affairs and deals with the river issues including environmental protection, water distribution, wastewater management, respect of environmental policies or laws, land and resource utilization, organization between stakeholders, etc., as shown in Figure 1.
Point 17: L68: I don't agree. They said "should be able" to achieve environmental goals, but it does not work. "should be able" is absolutely not "were able".
You should come back to the very good paper of Huang and Xu (2019) to precise what is the fundamental goal of RCS by definition from the Chinese Law in 2016. And after, you should provide your questioning you would like to discuss in your paper.
Respond 17: Accepting the reviewer’s valuable comment, we have carefully read the paper of Huang and Xu and made modifications to the expression on Lines 97 to 102 in the revised version:
Huang & Xu, on the other hand, presented a negative view on the implementation of RCS as they postulated that factors like power distribution, hierarchical systems, and public involvement could not be appropriately matched and would lead to the failure of environmental governance. They argued that, as a relatively closed water environment management system, the RCS was established primarily to strengthen the government’s water management rather than to create a platform for diversified governance [14].
The questionings we would like to discuss in our paper have been added on Lines 108 to 118 in the section of Introduction after referring to the mentioned manuscript:
Since the release of ‘The Opinions on Full Implementation of the River Chief System Across the Country’ in December 2016, as mentioned above, the RCS has been undergoing development nationwide. The main concerns of the current manuscript are as follows: (1) What are the recent developments under the institutional framework of the RCS? (2) What is the effectiveness of the RCS in preventing water pollution? (3) What are the weaknesses of the RCS and how might we facilitate its improvement? To address to these questions, we have summarized the latest developments in the system, in technical strategies of implementation, in the legislation of the RCS, and in the public outreach. The changes in the water quality of important watersheds in China over the past several years were also evaluated. By describing current developmental progress and outstanding weaknesses and problems of the RCS, this paper proposes some practical approaches by which the innovative development of the RCS can continue.
Point 18: L76-81: I am not sure it is the good place. I will put it in "materials and methods" (just before your line 92). The introduction should raise your questioning.
Respond 18: Modification has been made as suggested.
Point 19: L83-84: should be related to line 44.
Respond 19: Modifications have been made on Lines 108 to 110 in the revised version:
Since the release of ‘The Opinions on Full Implementation of the River Chief System Across the Country’ in December 2016, as mentioned above, the RCS has been undergoing development nationwide.
Point 20: L89-90: Comments on your introduction
So, to achieve this, you should use your introduction to highlight all the positive and negative reactions provided by the RCS experience in China. Maybe you could use more the paper of Huang and Xu (2019) which is very negatively critical of this system. They said:
"RCS was established mainly to strengthen the government’s water
management rather than to create a platform for diversified governance" and concluded on: "which might become an obstacle to effective water governance."
Here, you have a marvelous disruptive statement you could discuss.
Respond 20: We appreciate the reviewer’s valuable comment. More introductions on the positive and negative reactions provided by the RCS have been added in Introduction and in other parts of the manuscript.
Point 21: Materials and Methods
L92: I'll put here your lines 76-82
Respond 21: Modifications have been made as suggested.
Point 22: L92: precise what kind of data ?
Respond 22: Modifications have been on Lines 127 to 129 in the revised version:
Water quality data have been collected from the national bulletins and weekly reports from the Ministry of Water Resources and the China National Environmental Monitoring Centre [16,17].
Point 23: L103: so in your line 47, it would be better to emphasize this: you comment the RCS experience in China from last 3 years. It's clear and additional work for the international community.
Respond 23: Accepting the reviewer’s suggestion, modifications have been made on Lines 163 to 165 in the revised version to avoid misunderstanding:
The establishment of the document signified that the RCS had rapidly transitioned from an emergent policy to a nationwide action.
Point 24: Results
L160: I didn't understand the legend?
And with this figure, it seems that the RCS and LCS networks are not applied nationwide, could you explain?
Respond 24: We appreciate your valuable comment. As mentioned in the manuscript, the General Office of the CCP Central Committee and the State Council released the document ‘The Opinions on Full Implementation of the River Chief System (RCS) Across the Country’. By the end of 2018, a total of 24,000 hierarchical lake chiefs have been nominated at the provincial, city, county, and township levels across the nation. However, the application of the system has remained only at the administrative level, and does not have the legal effects, systematic and standardized. We have highlighted in the beginning of current section 3.1.4 that ‘Doubts regarding the RCS and LCS have long existed, a common criticism being that the system lacks the necessary legal strength, and that it will not last long-term.’ By performing legislations on RCS and LCS, the system will be legally enforceable. In the current section 3.1.4, we mainly focused on the recent legislations of RCS and LCS in certain provinces, which were the extension of the development of RCS/LCS since the release of the administrative document.
The legend has been revised to make more clarity:
Figure 2. The legislation process of river/lake chief systems at national level and provincial level since 2016. (Source: Official websites of each provincial government)
Point 25: It is apply only on the East border of China ?
Respond 25: Yes, the legislation of RCS and LCS was mainly conducted in the east border of China. But the system has been fully implemented across the nation administratively by the end of 2018 as mentioned above.
Point 26: L165: so, what about Jiansu (cf line 39) ?
Respond 26: Jiangsu has firstly implemented RCS administratively since 2012 as mentioned on line 39, but no specific legislation on RCS has been performed. While Zhejiang is the first province releasing a special legislation to implement the system.
Point 27: L166: maybe the figure 3 could listed these 19 clauses.
Respond 27: We appreciate the comment. Nevertheless, the contents of the 19 clauses were too long and complicated to be clearly shown in only one Figure. Consequently, we did not list these clauses as suggested.
Point 28: L168: So Fig 2 and 3 are only for 5 provinces?? Then you should adapt their title in function. Which province was the first? It’s not clear between line 39 with Jiangsu and line 165 with Zhejiang, I quote you: “The Regulations for RCS in Zhejiang Province’ was the first such special legislation” ??
Respond 28: We suppose the misunderstanding of the reviewer might be generated from the improper description of the legislation of RCS. As mentioned above, the RCS has been implemented across the nation administratively since 2016. But during the implementation process, the lack of financial support or improper might impede the effective implementation of RCS. Thus a legislation on RCS and LCS might enforce the system legally. Jiangsu was true the first to apply RCS across the province, but did not enact a specific legislation. In comparison, Zhejiang was the first to establish a special legislation for RCS.
Point 29: L170: I didn't understand what you mean? Explain
Respond 29: The purpose the authors wish to deliver here is as follows. As discussed in the front part of the paragraph, there are two ways to legislate for the RCS, in which the first is to establish a special legislation for the implementation of the system. For example, Zhejiang was the first to establish a special legislation for RCS. The second way to make RCS legal is to attach collateral provisions which associate the RCS with existing local laws or regulations targeting river/lake management, water pollution control, or water resource management. In some provinces like Jiangsu, some provisions or clauses associated with the RCS issues have been added in certain laws or regulations. To make the expressions with more clarity, the related sentences have been revised on Lines 178 to 181 as follows:
In some other provinces, mostly near the east border of China as shown in Figure 2, no specific legislation on the RCS has been established. Instead, these provinces have attached collateral provisions which associate the RCS with existing local laws or regulations targeting river/lake management, water pollution control, or water resource management.
Point 30: L176: again, I didn't understand the title of the figure with the legend text inside the figure. It seems to be more a calendar of RCS legislative decrees.
Respond 30: The title of the figure has been revised as the reviewer suggested.
Figure 3. The calendar of RCS legislative decrees since 2017 (Source: Official websites of each provincial government)
Point 31: L177: so, it is not a result but your statement that could be in the discussion part ?
Respond 31: Accepting the reviewer’s suggestion, the related statements have been moved to the discussion part.
Point 32: L179: you mean arrangements, facilities? by supplementary laws or decrees ?
Respond 32: Yes, the expressions have been revised on Lines 445 to 449 as follows:
At the national level, it has been suggested that the government establish special legislation for the RCS by legislation supplementary laws or decrees. More provisions regarding the scope of application, the delegation of responsibility, and the legal liability of river chiefs should be established. As for the detailed responsibilities of the local hierarchy of chiefs, these will need to be stipulated in local legislation.
Point 33: L181: I got some difficulties with this paragraph. Did you express here your own perception? Then it would be in the discussion part.
Or did you express a common evaluation of RCS already published? Then you should provide some citations.
Respond 33: Accepting the reviewer’s suggestion, the related statements have been moved to the discussion part.
Point 34: L185: you mean citizens?
Respond 34: Yes, modification has been made as suggested.
Point 35: L205: areas
Respond 35: Modification has been made as suggested.
Point 36: L208: you mean nominated, so designed by who? and are they paid? or any other advantages ? something else ?
Respond 36: The nomination was performed by local governments at province, city, county, and township levels, in which some are social volunteers and some are public servants. There is no accurate information whether they were paid or not, but most possibly not paid. The related expressions have been revised on Lines 314 to 316 as follows:
As of July 2018, more than 760,000 social river chiefs (including river cruisers and 208 river guards) at the province, city, county, and township levels had been nominated by local governments [25].
Point 37: Comments: This part could be shortened with clear details on what were the first clauses at the beginning and then what were the addition. Your current text is too long and not enough organized.
Respond 37: The paper has been re-organized as suggested.
Point 38: You should also provide the definition of a “river” and a “lake” for the RCS and LCS network: for example, which size ? There is a minimum amount of water provision ? Or is it related to the number of users ? or…????
Respond 38: Accepting the reviewer’s comment, the definitions of a river or a lake in this manuscript has been added on Lines 155 to 159 and on Line 180, in the revised version:
Based on the practical experience by local governments concerning the unique environmental, hydrological, and social conditions, a river with a basin area larger than 10 km2 is suggested to be included in the scope of RCS network. While smaller rivers or ditches with basin areas less than 10 km2 are to be merged into the management of the upper level rivers.
Generally, LCS network covers the management of lakes with areas larger than 1 km2.
Point 39: L216-218: What is the link with what you already said in lines 109-112 with citation 16: "On November 20, 2017, the General Office of the CCP Central Committee and the State Council released the document of ‘Guiding Opinions on the Implementation of the Lake Chief System (LCS) in Lakes’, and a general plan was made for the comprehensive implementation of both the RCS and LCS. On October 9, 2018, the Ministry of Water Resources issued the document ‘The Opinions on the Promotion of the System from ‘In Name’ to ‘In Practice’ "
Respond 39: There is a progressive relationship between the two opinions. The opinion released in 2017 was the first document aiming at implementing LCS. During the implementation process, the government has been encountering problems and difficulties. To solve the existing problems and to achieve a better performance of the LCS, ‘The Opinions on the Promotion of the System from ‘In Name’ to ‘In Practice’’ has been released with more specific regulations and guidance.
Point 40: L219: so, it would be more comprehensive that you develop this part before your description of the complements done by some specific provinces, such as discussed from line 162 to 209.
Respond 40: We appreciate the reviewer’s valuable comment. Section 3.2 was established mainly focusing on discussing the implementation strategies of RCS and LCS in a technical point of view. We have re-arranged and organized the major points during the implementation of RCS, and the efforts could be clearly observed by the reviewer.
Point 41: L223: you mean one document for one water body? so each considered river and each considered lake have each one one document?
Respond 41: Yes, it means one document for one water body here. To avoid misunderstanding, the expression has been revised to ‘One River (or Lake), One Document’ in the manuscript.
Point 42: L237: the quality parameters should be listed exhaustively. Then what about heavy metals ? POPs ?
Respond 42: Accepting the reviewer’s comment, more quality parameters have been listed on Lines 220 to 223 in the revised version:
Water quality parameters, such as chemical oxygen demand (COD), total nitrogen (TN), ammonia nitrogen (NH3-N), dissolved oxygen (DO), total phosphorus (TP), heavy metals, pH, turbidity, suspended solids, temperature, etc., have long served as the dominant factors used for evaluating a river’s status.
Since the concentrations of POPs in common rivers are not as high as in chemical-contaminated sites such as refuse landfills, they have not been applied and measured for evaluating a river’s status.
Point 43: L239: who pay to get these expensive paramters to be measured ??
Respond 43: We politely disagree with the reviewer that these parameters are expensive to be measured. Prior to evaluating the status of a river or lake according to the ‘One River (or Lake), One Document’, these parameters are to be measured by the Office of the River Chief System, or by professional organizations in water environment assessments.
Point 44: L242: explain?
Respond 44: The authors intended to highlight that the conventional application of these water quality parameters could not comprehensively evaluate the situations of water environment and water ecology. The related expressions have been revised on Lines 229 to 231 in the revised version as follows:
Nevertheless, these parameters only partially reflect the quality of water body, which do not fully characterize the dynamic changes in water environments and water ecology.
Point 45: L244-250: In fact, it is not so simple to qualify the quality of the waters. It depends about the final use. So the few explanations provided here are not satisfying. Since it is not the core of this paper, it would be better to avoid this issue. Just you explain in few words that the RCS and LCS policy to characterize the water quality.
Respond 35: Accepting the reviewer’s comment, the aforementioned section has been revised on Lines 231 to 235 in the revised version as follows:
Consequently, the ‘One River (or Lake), One Document’ suggests that in addition to the conventional water quality parameters, evaluations of the ecological health of rivers are also required. For example, the parameters of the index of biotic integrity, or microbial index of biotic integrity are worth utilizing to provide both technical and theoretical support for the accurate assessment of river health [27-31].
Point 46: The papers 29 to 31 should/could be more exploited and described to legitimate the use of this index in your paper. But it's not this index you plot in figures 6 to 9, why?
Respond 46: We appreciate your valuable comment. The authors intended to indicate that the current survey or evaluation of rivers or lakes lack some biological or ecological parameters, which needs to be concerned and covered in the establishment of ‘One River (or Lake), One Document’ in the future.
Point 47: L273: oh! Are they the main pollutants? What about the trace elements such as A, Cr, B (Xiao, Wang et al, 2019, STOTEN), metal pollution (Zhang, Qin et al, 2018, STOTEN) and microplastics (Wang, Ndungu, 2017, STOTEN), and so on...??
Respond 47: We appreciate the reviewer’s valuable comment. According to the ‘National Water Resources Bulletin’ released by the Ministry of Water Resources annually, the concerned pollutants were conventional parameters such as TN, TP, NH3-N and COD etc. While the pollutants such as metals and microplastics were not among the concerned pollutants from the data source, and no specific description on these pollutants were provided in the manuscript.
Point 48: L286-288: I didn't understand: it is the opposite of what you say Line 275! And I agree with the statement of the line 275.
Respond 48: Accepting the reviewer’s suggestion, the related expressions have been revised on Lines 337 to 346 in the revised version with more clarity:
Figure 6 illustrates the annual data on the number of rivers or lakes that attained each water quality standard from 2015 to 2019. The number of rivers with water quality worse than Grade V has gradually decreased since 2015 as shown in Figure 6(a). In comparison, the number of rivers meeting Grade I to Grade II standards has increased. Figure 6(b) indicates that the number of key lakes meeting Grade I and Grade II standards also increased in the past years, but no clear changes were observed in heavily-polluted lakes. These results may indicate that the implementation of the RCS has increased the proportion of water bodies that have high water quality, i.e., Grade II or better. However, the RCS does not appear as effective in eliminating pollution from heavily polluted lakes. The long-term effects of the RCS on river and lake health are still unknown and need to be monitored further.
Point 49: However if you really want to prove the efficiency of RCS&LCS on water quality, you should provide data. But I didn't think its the aim if this paper, then I would delete this paragraph (from line 268).
Respond 49: We politely disagree with the reviewer to delete the paragraph. In this section, we mainly focused on the water quality promotion in representative basin and lake in China. The results provide additional evidence that the implementation of RCS aided in the water quality improvements.
Point 50: L289: oh, it should be reported inside the method paragraph.
Respond 50: Accepting the valuable comment, the expressions have been added to the method paragraph.
Point 51: L294: you cannot be so affirmative. Moreover in this result part, you should just describe the data, and discuss on them!
Respond 51: We appreciate the reviewer’s comment. We have deleted the discussion part as suggested.
Point 52: BTW, it would be good to present these 2 graphs with the STD values. And more than the % the numbers of water bodies would be better (number of rivers, numbers of lakes). You should describe that in general there are no significant evolution and sometimes more degradation.
Respond 52: We have checked the raw data and there were no repetitive data on the number of rivers or lakes reaching each water quality standard. Thus STD values could not be obtained in these graphs.
Accepting the reviewer’s suggestion, the number of rivers were added in the figure in replacement of percentage values.
(a) |
(b) |
Figure 6. Water quality of (a) 978 of the major rivers including the seven major river systems, and (b) 112 major lakes, including Taihu Lake, Dian Lake, etc., in China. Data source: Ministry of Ecology and Environment of the People’s Republic of China
Point 53: L295: I don't understand the two vertical legends of your 2 graphs. For b, you mean: "Number of lakes ..." and not rivers ?
Why you used "Env quality standard" for a, and "degree of contamination" for b ?
I would write "Env quality standard" for both.
Respond 53: We appreciate the reviewer’s valuable comment. We have stressed in the legend of Figure 6 that Figure 6 (a) refers to the water quality of the major rivers including the seven major river systems. While Figure 6(b) refers to 112 major lakes including Taihu Lake, Dian Lake, etc. in China.
For the interpretation of the vertical legends, the expression of “degree of contamination” was referred from the raw data source of the Ministry of Ecology and Environment of the People’s Republic of China. The ministry has defined two separate standards for defining the water qualities of rivers and lakes, respectively. We have modified Figure 6(b) as suggested as in response to the above comment.
Point 54: Comments:It could be interesting to show these data. So you should describe how did you do to calculate the index of quality for rivers and lakes, to describe also how many measurements per station, etc…
Respond 54: We appreciate the reviewer’s valuable comment. Actually we did not calculate the index of quality of rivers and lakes here, all the data were referred from the source website. We have surveyed the website and no explanation on the number of measurements per station could be found.
Point 55: L303: oh no ! you cannot compare interannual data and here intermonthly data. The evolution along the year could be explained by the climatic seasons, by the cycle of human activities, by the economic conjonctures, etc...
Respond 55: We appreciate the reviewer’s valuable comment. But we did not compare the interannual data with intermonthly data here. We just compared the situation in September, 2016 and January, 2020, which were all intermonthly data. Accepting the reviewer’s valuable comment, we have re-collected the annual data and modified Figures 6(a) and 6(b) as suggested.
Point 56: Then this paragraph should be deleted.
Respond 56: We politely disagree with the reviewer deleting the paragraph. In this paragraph, we have collected data expressing the water quality changes in the biggest river basin, Yangtze River Basin, which provided supportive evidence on the performance of RCS implementation.
Point 57: L328: explain this concept in your text
Respond 57: Explanation has been provided on Lines 375 to 377 in the revised version as suggested.
A water function zone refers to an area that has a specific function, and has met the requirements of rational development, utilization, and protection of water resources in accordance with the regional water resources development and social needs.
Point 58: L331: meaning? is it upstream rivers?
Respond 58: The caption of Figure 9 has been revised to make more clarity:
Figure 9. The proportions of upstream rivers with water quality superior to Grade III from 2007 to 2018 in Taihu Basin.
Point 59: Comments: If you want to describe some specific cases to demonstrate the impacts of the RCS&LCS, then you should describe the study cases before. Without that, it is difficult to be convinced.
Respond 59: We appreciate the reviewer’s valuable comment. In this section, we have mainly focused on the water quality change improvements in the past three years since the full implementation of RCS and LCS in 2016. The reviewer is right that the improvements could not be directly demonstrated to be impacted by the implementation of RCS&LCS. Accepting the reviewer’s suggestion, we have revised some expressions on Lines 337 to 346 in the manuscript to make discussion with more clarity:
Figure 6 illustrates the annual data on the number of rivers or lakes that attained each water quality standard from 2015 to 2019. The number of rivers with water quality worse than Grade V has gradually decreased since 2015 as shown in Figure 6(a). In comparison, the number of rivers meeting Grade I to Grade II standards has increased. Figure 6(b) indicates that the number of key lakes meeting Grade I and Grade II standards also increased in the past years, but no clear changes were observed in heavily-polluted lakes. These results may indicate that the implementation of the RCS has increased the proportion of water bodies that have high water quality, i.e., Grade II or better. However, the RCS does not appear as effective in eliminating pollution from heavily polluted lakes. The long-term effects of the RCS on river and lake health are still unknown and need to be monitored further.
Point 60: Discussion: L345: but Huang and Xu, 2019, had the opposite discours.
Respond 60: We appreciate the comment. In the opinion of the authors, the RCS has overcome difficulties in regional cooperation and responsibility distribution among different departments, and we have drawn to the conclusion that RCS is beneficial in watershed managements throughout the whole manuscript.
Point 61: L347: innovative for China maybe, but not with other integrated river management systems in the world.
Respond 61: Appreciating the valuable comment, the sentence has been removed and more discussion on the superiority and necessity of RCS has been added in section 4.1:
From the perspective of the top-down hierarchy, the river and lake chiefs are imbued with a strong sense of regional cooperation, which is conducive to increasing the efficiency of water resource management and ecological protection [4]. Under the guidance of water pollution control plan, effective river/lake management will be achieved by breaking up the goals into manageable pieces and hierarchically delegating the targets. Moreover, strict evaluation and assessment will be integrated to track the chiefs and ensure they are efficient in achieving their target goals. Tackling the existing problems with Chinese watershed management, the RCS system sorted out the collaborative problems in water management among departments, local government, and various political levels.
Point 62: L352: then it is not done yet, isn't it?
Respond 62: No, it isn’t. According to the rules and opinions released by the General Office of the CCP Central Committee, the State Council or the Ministry of Water Resources, the river/lake managements should have been achieved by decomposing the goals and hierarchically transferring the targets. Nevertheless, during the implementation process, the management targets could not always be achieved due to the unclear duties and responsibilities of river chiefs and the incomplete accountability and supervisory systems as discussed in the following section in the manuscript.
Point 63: So the RCS&LCS network is not entirely set up ?
Respond 63: Yes, as mentioned above, the system has been fully implemented by the end of 2018. However, during the process, still a lot of problems exist and the national and local governments have been attempting to optimize and strengthen the implementation. On October 9, 2018, the Ministry of Water Resources issued the document ‘The Opinions on the Promotion of the System from ‘In Name’ to ‘In Practice’.
Point 64: L362: It would be very interesting to discuss that point. is it enough? What about the task of water distribution? water allocation?
Respond 64: The related expressions have been revised on Lines 419 to 426 in the revised version as follows:
The action organically combined the tasks of channel improvements, flood control, water pollution control, environmental protection, ecological remediation, water allocation, and water conservancy. In this way, a water management approach with clear basis in traditional Chinese top-down hierarchy can be achieved. In a case study performed by Liu et al. where they surveyed the outcomes of the RCS in Foshan, China, the increased rate of rivers meeting water quality goals after implementation of the RCS clearly demonstrated the validity of the collaborative system in river pollution control, water conservancy, and ecological restoration [13].
Point 65: L373: is it a necessity ??? you should read some papers from the outside China.
Respond 65: Accepting the reviewer’s valuable comment, the related expressions have been deleted.
Point 66: L337: where? how? by who? Your sentence is an assertion.
Respond 66: We appreciate the valuable comment. We have revised the related expressions on Lines 390 to 396 as suggested:
As a response to the serious water environment issues by a public governance system, the RCS offered a significantly superior approach that has since become an indispensable tool in Chinese watershed management. River chiefs at all administrative levels are responsible for the supervision and management of the rivers and lakes, including protection of water resources, coastline management, water pollution control, water environmental management, leading the organization of the occupation of rivers, participating in the reclamation of lakes, and monitoring of excessive sewage, illegal sand mining, destruction of waterways, illegal fish hunting, etc.
Point 67: L391: always of nowadays? or it was before?
If it was before, write: were.
Respond 67: Change has been made on the expression on Lines 452 to 454 as suggested:
During the implementation of the river/lake chief systems, problems such as unclear distributions of responsibilities, unclear authority, and poor interoperability persist.
Point 68: L400: and not by being multi-actors ? It is here the key question of this paper.
Respond 68: It’s not by being multi-actors.
Point 69: L402: no, not "can" but "should". It is obliged to have many changes between upstream and downstream at each environmental organization level. Etc...
Respond 69: Modification has been made as suggested.
Point 70: L403-405: YES
L407: oh ! I am surprised since I thought that RCS/LCS systems were to avoid that issue.
Respond 70: We appreciate the valuable comment. The RCS/LCS system is designed aiming to solve the problems in lacking comprehensive governance and trans-regional cooperation. Nevertheless, during the implementation, the management targets could not always be achieved due to the limited governance and management.
Point 71: L412: Yes, it is exactly what your paper should discuss. How to do it ? your ideas ? the requirements ?
Respond 71: More discussion on this topic has been added on Lines 428 to 449 in the revised version:
Since the full implementation of the RCS in 2016, the management system and working hierarchy have been substantially improved. However, there have been a variety of problems that this system has encountered, with numerous problems arising early during its implementation. These problems stem from three major sources. The first being that, in many cases, the main duties and responsibilities of river chiefs remain unclear. As mentioned above, while the amount of legislation on the river/lake chief systems has accelerated in the past three years, and in light of the release ‘Law of the People's Republic of China on Prevention and Control of Water Pollution’ and the numerous local regulations and normative documents, legal provisions on the statutory duties of river chiefs remain limited. The duties of river/lake chiefs, and other supervisory departments, have not been well defined and delegated, and the ‘rule by man’ approach can be seen in the real-world applications. Critics also argue that the overreliance on the personal power of river chiefs can cause severe responsibility gaps in the implementation of the RCS [4]. In the critical review by Huang & Xu, the authors argue that the strengthening of local authority could enable local river chiefs to combat or eliminate state power [14]. The authors point out that the RCS presents characteristics of a bureaucratic environmental governance, which has previously become an obstacle to effective water governance [14]. Consequently, legislation on the RCS is essential to define clear legal responsibilities of river chiefs at different administrative levels. It is foreseeable that along with the development of the RCS, legislation for the RCS will be needed nationwide. At the national level, it has been suggested that the government establish special legislation for the RCS by legislation supplementary laws or decrees. More provisions regarding the scope of application, the delegation of responsibility, and the legal liability of river chiefs should be established. As for the detailed responsibilities of the local hierarchy of chiefs, these will need to be stipulated in local legislation.
Point 72: L413: explain
Respond 72: Comprehensive governance here means integrated management of watersheds with coordinated systems, broad communication channel and sound public participation.
Point 73: L419: and what about the individual needs and wishes?
Respond 73: Accepting the reviewer’s suggestion, individual needs and wishes have been added on Lines 493 to 497 in the related section:
A multi-channel guarantee for the realization of the river/lake chief system needs to be established from the perspectives of law, policy, management, technology, informatization, and public participation. The final task will be to set up a long-term government-led river management mechanism based on clear rules with participation from various sources including the public, the media, and individuals.
Point 74: Conclusion
L423: not really
Respond 74: We have re-organized the result sections to make the interpretation with more clarity.
Point 75: L424: yes, but should be better organized.
Respond 31: We appreciate the valuable comment and have re-organized the related sections in the manuscript.
Point 70: L425: why this sentence to emphasize the LCS? It is not what we have inside the text.
Respond 75: We think the confusing of the reviewer is largely due to the misreading of section ‘3.1.1 The implementation of the Lake Chief System (LCS)’, in which we have highlighted the emergence and novelty of LCS. LCS is originated from RCS, but different with RCS in terms of the objects and targets of management. It is one of the main development achievements in the past several years and we have made a emphasize on the LCS, as the reviewer asked.
Point 76: L426: it is exactly what your text should develop. Maybe you could organize your paragraphs around this concern:
- RCS related to national, regional scales
- RCS related to local, individual requirements
Respond 76: We appreciate the reviewer’s valuable comment. We have read the very important paper ‘River-basin planning and management: The social life of a concept, Geofrum, 2009’ as recommended by the reviewer. I suppose the reviewer is suggesting us to discuss how the RCS has been associated with national, regional scales and what are the local and individual requirements for RCS. Nevertheless, owing to the limited understanding and knowledge on social sciences, we feel difficult to discuss the related issues with clear viewpoints and statements.
Point 77: L430: absolutely not! You cannot write a so affirmative result.
Respond 77: Accepting the reviewer’s suggestion, modification has been made on Lines 510 to 513 in the revised version:
The collected data have shown that after the full implementation of the RCS, water quality nationwide has improved, especially terms of the proportion of water bodies that meet high water quality standards and in eliminating the existence of heavily-polluted rivers.
Point 78: L437-440: It is exactly what we would like to be demonstrated in the paper. But it is not done yet.
Respond 78: We appreciate the reviewer’s comment. More discussions and expressions related to the unresolved problems have been added on Lines 428 to 481 in the revised version. Indeed, we have made great efforts to improve the manuscript to address the reviewer’s concerns.
Since the full implementation of the RCS in 2016, the management system and working hierarchy have been substantially improved. However, there have been a variety of problems that this system has encountered, with numerous problems arising early during its implementation. These problems stem from three major sources. The first being that, in many cases, the main duties and responsibilities of river chiefs remain unclear. As mentioned above, while the amount of legislation on the river/lake chief systems has accelerated in the past three years, and in light of the release ‘Law of the People's Republic of China on Prevention and Control of Water Pollution’ and the numerous local regulations and normative documents, legal provisions on the statutory duties of river chiefs remain limited. The duties of river/lake chiefs, and other supervisory departments, have not been well defined and delegated, and the ‘rule by man’ approach can be seen in the real-world applications. Critics also argue that the overreliance on the personal power of river chiefs can cause severe responsibility gaps in the implementation of the RCS [4]. In the critical review by Huang & Xu, the authors argue that the strengthening of local authority could enable local river chiefs to combat or eliminate state power [14]. The authors point out that the RCS presents characteristics of a bureaucratic environmental governance, which has previously become an obstacle to effective water governance [14]. Consequently, legislation on the RCS is essential to define clear legal responsibilities of river chiefs at different administrative levels. It is foreseeable that along with the development of the RCS, legislation for the RCS will be needed nationwide. At the national level, it has been suggested that the government establish special legislation for the RCS by legislation supplementary laws or decrees. More provisions regarding the scope of application, the delegation of responsibility, and the legal liability of river chiefs should be established. As for the detailed responsibilities of the local hierarchy of chiefs, these will need to be stipulated in local legislation.
Second, the accountability and supervisory systems remain incomplete. The RCS aims to solve the key issue of assigning responsibility for specific water affairs. Along with this, the need for specific evaluation modes and methods for accountability has been identified. During the implementation of the river/lake chief systems, problems such as unclear distributions of responsibilities, unclear authority, and poor interoperability persist. The practice of shirking duties and responsibilities might occur more frequently if there is no clear system of accountability. In practice, the impartiality of an accountability review process may not be guaranteed because environmental bureaus or RCSO, which are responsible for the implementation of the process, are lower on the hierarchy than the river chiefs at administrative levels [40]. The accountability of river chiefs is imposed within the party-state hierarchy, by upper-level government, and is therefore susceptible to rent-seeking and corruption. According to the field investigations carried out by Wang & Chen, no river chiefs have ever been relieved of their roles for poor management under the ‘One-Vote Veto System’ [4]. The current accountability mechanisms lack legal foundation. Additionally, the supervision and management mechanisms of the RCS are not perfect. As a result, there is a proportion of river chiefs that exist in name only, having no assigned tasks or responsibilities.
Third, the management of rivers and lakes has not been integrated at a large scale, with no clear comprehensive plan or coordination. The goals of the RCS are typically characterized by being trans-basin and trans-regional. However, where the RCS has been implemented on cross-border rivers, the administrative management standards and requirements sometimes differ between upstream and downstream areas, sometimes even on different sides of the same river. The effects of the differences between management on the upper reaches and lower reaches are difficult to quantify, which consequently leads to difficulty in assigning responsibility and further complicates river management. Specifically, the implementation of the RCS often lacks unification and coordination across multiple regions. The management of rivers and lakes is often unilateral, and comprehensive governance at the regional level is still lacking. The complexity of transboundary water issues has puzzled local governments as was shown in an important article surveying the science and negotiation theory of resolving boundary-crossing water issues. The article by Islam & Susskind called for the negotiated choice of decomposition among different regions, which has become an essential component of understanding, governing, and managing complex water systems [41]. An additional concern of the widespread deployment of the RCS is the variable financial support from different governments. Compared to the more developed eastern border of the nation, it may be more difficult for less developed regions to provide sufficient support for effective water governance.
Point 79: Resume
L17 : ou have listed the actions done to implement the RCS. But you didn't analyze the implementation actions. It would be very interesting to do it.
Respond 79: The implementation actions have been added by referring to some important researches as suggested.
Point 80: L23 : No, your paper is very far away to prove that assertion.
Respond 80: Modification has been made on Lines 20 to 24 in the revised version as suggested.
Finally, a summary of the weaknesses and outstanding problems of the system is presented, putting forward a multi-channel strategy for the long-term stability and effectiveness of river/lake chiefs, and promoting the RCS as a suitable solution to the collaborative and jurisdictional issues in water management in China.
Reviewer 2 Report
Dear Authors,
Thank you very much for this interesting work
In case that the editor ask a revised manuscript,
i am happy to serve as reviewer of the revised manuscript
Kind Regards,
Reviewer
Author Response
We appreciate the comment from the reviewer. This manuscript has been carefully revised according to the constructive suggestions from the three reviewers. For each point raised, a detailed explanation and a general description of the revisions have been provided.
Reviewer 3 Report
The manuscript sounds interesting to the readers and presents a good approach to manage rivers/lakes. Please check my comments to enhance your submission.
- Introduction: It seems that you missed references 8 through 11, please check.
- Introduction: In line 60, Please change the caption of figure 1 into “Schematic diagram of the river chief system (RCS)”
- Introduction: In line 72, “Wang et al. was included as a reference No. 4. That’s correct, buy that reference includes only two authors. So, it is more acceptable to update your in-text citation into Wang & Chen.
- Introduction: In line 77-78, remove quotations from ‘Research……….University’.
- Introduction: In line 81, “at the time of writing”. Please make it more clear to the readers what you were writing.
- Introduction: In line 90, please change “Healthy” into innovative.
- Introduction, it seems you have discussed all advantages of RCS. However, potential disadvantages of RCS should also be included.
- Section 2: Grammatical errors were noticed in more than one occasion. Please revise.
- Conclusion: In line 429, change “Water qualities” to water quality. Use water quality instead of water qualities throughout the article.
- Conclusion: In line 435, “and in doing has provided….”. It is unclear what you were trying to let the readers be aware of.
- Conclusion: In line 438-440, you pointed out to some improvements in the RCS. This discussion should be explained and included in introduction. Please refer to my comment # 6.
- Results: In line 112 & line 118, delete “the document”
- Sub-section 3.1.1: Line 127, correct “an important part” to important parts.
- Sub-section 3.1.1: Line 128 and everywhere in the article, change km2 to km2.
- Sub-section 3.1.2: Line 172, change 4 to four. Note that numbers from 1-9 should be spelled out in the text.
- Figure 3: Please include a reference in the caption if applicable.
- Sub-section 3.2.1: In lines 222-227, re-write for more clarity. Try to shorten sentences to minimize grammatic errors.
- Figures 4 & 5: add a reference in the caption.
- Sub-section 3.2.1: In lines 240-242, what are those methods that you mentioned used for? Add to the text for more clarity and connection with water quality parameters.
- Sub-section 3.3.1: In lines 289-294, why you selected annual water quality parameters for the months of September and January? Does water quality parameters during the other months of the year show the same trend. I suggest adding another analysis in the text.
- Sub-section 3.3.1: In lines 291-292, change figure 6A and figure 6B into figure 6 (a) and figure 6 (b).
- Sub-section 3.3.2: In lines 300-301, you mentioned “we found that the overall water quality was good in the Yangtze River basin”. Based on theory or standards you drew your finding? Add more clarity to the text.
- Sub-section 3.3.1: In line 302, change “water qualities” to water quality and use water quality elsewhere in the text.
- Figure 7 was not referenced in the text. Add a reference of figure 7 to the text in subsection 3.3.1.
- Figure 9: Explain what happened in 2013 caused the curve to reach a minimal percentage.
- Section 4.2: In line 389, delete “to” in (to completed).
- Overall, check all in-text references and make sure they are included in the reference list and vice versa. Also,
Author Response
Response to Reviewer 3 Comments
Point 1: The manuscript sounds interesting to the readers and presents a good approach to manage rivers/lakes. Please check my comments to enhance your submission.
Response 1: This manuscript has been carefully revised according to the constructive suggestions from the three reviewers. For each point raised, a detailed explanation and a general description of the revisions have been provided. The main modifications in manuscript text have been explained, and all the changes in the manuscript have been highlighted.
Point 2: Introduction: It seems that you missed references 8 through 11, please check.
Response 2: We appreciate the comment. All the references have been checked and revised as suggested.
Point 3: Introduction: In line 60, Please change the caption of figure 1 into “Schematic diagram of the river chief system (RCS)”
Response 3: The caption has been revised as accepted.
Point 4: Introduction: In line 72, “Wang et al. was included as a reference No. 4. That’s correct, buy that reference includes only two authors. So, it is more acceptable to update your in-text citation into Wang & Chen.
Response 4: The expression has been revised on Lines 104 to 107 in the revised version as suggested:
Wang & Chen analyzed the RCS by establishing an analytical framework according to collaborative governance theory, and concluded that the RCS was effective in tackling collaborative issues in water management, but they also noted its long-term impacts and sustainability cannot be accurately predicted
Point 5: Introduction: In line 77-78, remove quotations from ‘Research……….University’.
Response 5: The authors politely disagree removing the quotations. This is the full name of the first academic center entirely dedicated to RCS research across the nation, which the authors were employed in. The authors intended to highlight the importance of the institute here. We have moved the related expressions to the section of ‘Materials and Methods’ as reviewer 1 recommended.
Point 6: Introduction: In line 81, “at the time of writing”. Please make it more clear to the readers what you were writing.
Response 6: Modifications have been made on Lines 125 to 127 as suggested:
Note that, at the time of writing the current manuscript, the authors were employed by the research center and committed to the study of the RCS
Point 7: Introduction: In line 90, please change “Healthy” into innovative.
Response 7: Change has been made on Lines 116 to 118 in the revised version as suggested.
By describing current developmental progress and outstanding weaknesses and problems of the RCS, this paper proposes some practical approaches by which the innovative development of the RCS can continue.
Point 8: Introduction: it seems you have discussed all advantages of RCS. However, potential disadvantages of RCS should also be included.
Response 8: Appreciating the reviewer’s comment, potential disadvantages of RCS have been added on Lines 70 to 77, in the revised version:
As an emerging system in response to complicated water issues, the deployment of the RCS has, understandably, encountered problems and difficulties. The non-statutory responsibilities of river chiefs, the over reliance on administrative power, the trans-provincial management of rivers and lakes, as well as the lack of public participation and social supervision all may impede the effective implementation of the RCS. These potential impediments will be discussed in the following sections of the manuscript. Nonetheless, in recent years the RCS has proven itself as an effective method for solving complex collaborative problems in water management, which is deeply rooted in the country’s unique political systems [4].
Point 9: Section 2: Grammatical errors were noticed in more than one occasion. Please revise.
Response 9: The whole manuscript has been carefully revised by an English-speaking editor as the reviewer suggested. Indeed, we have made great efforts to improve the manuscript to address the reviewers’ concerns.
Point 10: Conclusion: In line 429, change “Water qualities” to water quality. Use water quality instead of water qualities throughout the article.
Response 10: Modifications have been made as suggested throughout the article.
Point 11: Conclusion: In line 435, “and in doing has provided….”. It is unclear what you were trying to let the readers be aware of.
Response 11: The expression has been modified on Lines 523 to 526 in the revised version as the reviewer suggested:
The RCS has undergone a migration to national governance and has expanded our knowledge regarding the diversity of choices in water management tools, providing insights that may prove useful to other developing countries seeking to establish a river management system.
Point 12: Conclusion: In line 438-440, you pointed out to some improvements in the RCS. This discussion should be explained and included in introduction. Please refer to my comment # 6.
Response 12: Related discussions on the improvements in the RCS have been added on Lines 66 to 77 in the revised version in the introduction section as suggested.
By resolving issues created by incomplete legal systems and insufficient judicial guarantees, this innovative system may be the answer to current, complicated water issues [12]. The main tasks of the chiefs include water resources protection, shoreline management, water pollution prevention and control, water environment management, restoration of water ecology, and law enforcement. As an emerging system in response to complicated water issues, the deployment of the RCS has, understandably, encountered problems and difficulties. The non-statutory responsibilities of river chiefs, the over reliance on administrative power, the trans-provincial management of rivers and lakes, as well as the lack of public participation and social supervision all may impede the effective implementation of the RCS. These potential impediments will be discussed in the following sections of the manuscript. Nonetheless, in recent years the RCS has proven itself as an effective method for solving complex collaborative problems in water management, which is deeply rooted in the country’s unique political systems [4].
Point 13: Results: In line 112 & line 118, delete “the document”
Response 13: Deletion has been made as suggested.
Point 14: Sub-section 3.1.1: Line 127, correct “an important part” to important parts.
Response 14: Correction has been made as suggested.
Point 15: Sub-section 3.1.1: Line 128 and everywhere in the article, change km2 to km2.
Response 15: Changes have been made as suggested.
Point 16: Sub-section 3.1.2: Line 172, change 4 to four. Note that numbers from 1-9 should be spelled out in the text.
Response 16: Modification has been made as suggested.
Point 17: Figure 3: Please include a reference in the caption if applicable.
Response 17: The caption of Figure 2 (since the structure of the manuscript has been changed as requested by reviewer 1, the previous Figure 3 transforms to be Figure 2 in the current version) has been modified as follows:
Figure 2. Legislation on river/lake chief systems at the provincial levels since 2017 (Source: Official websites of each provincial government)
Point 18: Sub-section 3.2.1: In lines 222-227, re-write for more clarity. Try to shorten sentences to minimize grammatic errors.
Response 18: Accepting the reviewer’s suggestion, the sentences have been re-written on Lines 205 to 210 in the revised version as follows:
‘One River (or Lake), One Document’ is a collection of information regarding the natural characteristics, development, management protection, and dynamic changes of rivers and lakes. The Ministry of Water Resources has issued ‘The Guide for ‘One River (or Lake), One Document’ Programming’ in April 2018 [25]. A document for any specific river or lake should contain information on the water intake, sewage discharge, water quality, water ecology, coastline development, river channel utilization, and water-related projects and facilities.
Point 19: Figures 4 & 5: add a reference in the caption.
Response 19: Reference has been added as suggested.
Point 20: Sub-section 3.2.1: In lines 240-242, what are those methods that you mentioned used for? Add to the text for more clarity and connection with water quality parameters.
Response 20: The application and purpose of these methods have been added on Lines 223 to 229 in the revised version:
Owing to the complicated hydrological conditions and lack of universal mathematical models for the evaluation of water quality, specialized evaluation approaches are usually required to evaluate the monitored water quality data from a specific area and determine the water quality categories and the spatial and temporal variations of water quality. These parameters are used in concert with methods such as the single-factor assessment method, water quality grading method, the Nemerow’s pollution index, comprehensive pollution index, principle component analysis, and the fuzzy comprehensive evaluation [26, 27].
Point 21: Sub-section 3.3.1: In lines 289-294, why you selected annual water quality parameters for the months of September and January? Does water quality parameters during the other months of the year show the same trend. I suggest adding another analysis in the text.
Response 21: We appreciate the reviewer’s valuable comment on the data selection. We have re-selected the annual water quality data from 2015 to 2019 to reflect the changes. The related expressions have been revised on Lines 337 to 349 as suggested.
Figure 6 illustrates the annual data on the number of rivers or lakes that attained each water quality standard from 2015 to 2019. The number of rivers with water quality worse than Grade V has gradually decreased since 2015 as shown in Figure 6(a). In comparison, the number of rivers meeting Grade I to Grade II standards has increased. Figure 6(b) indicates that the number of key lakes meeting Grade I and Grade II standards also increased in the past years, but no clear changes were observed in heavily-polluted lakes. These results may indicate that the implementation of the RCS has increased the proportion of water bodies that have high water quality, i.e., Grade II or better. However, the RCS does not appear as effective in eliminating pollution from heavily polluted lakes. The long-term effects of the RCS on river and lake health are still unknown and need to be monitored further.
(a) |
(b) |
Figure 6. Water quality of (a) 978 of the major rivers including the seven major river systems, and (b) 112 major lakes, including Taihu Lake, Dian Lake, etc., in China. Data source: Ministry of Ecology and Environment of the People’s Republic of China
Point 22: Sub-section 3.3.1: In lines 291-292, change figure 6A and figure 6B into figure 6 (a) and figure 6 (b).
Response 22: Changes have been made as suggested.
Point 23: Sub-section 3.3.2: In lines 300-301, you mentioned “we found that the overall water quality was good in the Yangtze River basin”. Based on theory or standards you drew your finding? Add more clarity to the text.
Response 23: Modifications have been made on Lines 352 to 354, in the revised version:
Based on ‘The weekly report on automatic water quality monitoring of major river basins and key sections of lakes and reservoirs in China’ from 2016 to 2019 [16,17], we found that the overall water quality was good in the Yangtze River basin.
Point 24: Sub-section 3.3.1: In line 302, change “water qualities” to water quality and use water quality elsewhere in the text.
Response 24: Modifications have been made as suggested.
Point 25: Figure 7 was not referenced in the text. Add a reference of figure 7 to the text in subsection 3.3.1.
Response 25: Reference of Figure 7 has been made as suggested on Lines to in the revised version:
Figure 7. Water quality in the Yangtze River basin [16, 17].
Point 26: Figure 9: Explain what happened in 2013 caused the curve to reach a minimal percentage.
Response 26: We appreciate the reviewer’s comment and have checked the raw data from the source website. However, no clear explanation for the minimal percentage in 2013 could be found from the source website. We believe that the upward trend of the proportions of upstream rivers with water quality meeting Grade III or better was clear in Figure 9. The lowest value in 2013 did not affect the interpretation of the results.
Point 27: Section 4.2: In line 389, delete “to” in (to completed).
Response 27: Modification of the sentence has been made on Line 450 as suggested.
Second, the accountability and supervisory systems remain incomplete.
Point 28: Overall, check all in-text references and make sure they are included in the reference list and vice versa. Also,
Response 28: Accepting the reviewer’s comment, we have checked all the references and reference lists in the text.
Round 2
Reviewer 1 Report
Dear Authors,
Thanks a lot for your comprehensive responses, very useful to understand your changes related to my comments.
And thanks for the new title : ‘Full implementation of the River Chief System in China: outcome and weakness’.
I am still not convinced by the use of the water quality data to prove the success of the RCS system. And I feel you don't focus enough on the importance of the local stakeholders, on the relationship between the RCS system and the entrepreneurial arena. But it's just my opinion and not yours. It should first your paper !
So, again, thanks a lot for your effort and I was really happy to read this new version.
Cheers
Reviewer 2 Report
Dear authors,
thank you very much for this interesting work
Best Regards,
Reviewer
Reviewer 3 Report
The manuscript has significantly been improved.